



# A revised northern soil Hg pool, based on western Siberia permafrost peat Hg and carbon observations

Artem G. LIM[1], MARTIN JISKRA[2], Jeroen E. SONKE[3],

Sergey V. LOIKO[1], NATALIA KOSYKH[4], Oleg S. POKROVSKY[3,5*]

[1] *BIO-GEO-CLIM Laboratory, Tomsk State University, Tomsk, Russia*
[2] *University of Basel, Environmental Geosciences, Bernoullistrasse 30, 4056 Basel, Switzerland*
[3] *Geosciences and Environment Toulouse, UMR 5563 CNRS, 14 Avenue Edouard Belin 31400 Toulouse, France*
[4] *Lab Biogeocenol, Inst Soil Science & Agrochem, Russian Acad Sci, Siberian Branch, Novosibirsk, Russia*
[5] *N. Laverov Federal Center for Integrated Arctic Research, Russian Academy of Sciences, Arkhangelsk, Russia*

*Email: oleg.pokrovsky@get.omp.eu*

*Key words: mercury, peat soil, landscape, bog, , forest, thaw, Siberia*

*Submitted to  Biogeosciences December 2019*





**Abstract**
Natural and anthropogenic mercury (Hg) emissions are sequestered in terrestrial soils over short,
annual, to long, millennial time scales, before Hg mobilization and run-off impacts wetland and
coastal Ocean ecosystems. Recent studies have used Hg to carbon (C) ratios, $R_{HgC}$, measured in
Alaskan permafrost mineral and peat soils, together with a northern soil carbon inventory, to
estimate that these soils contain large amounts, 184 to 755 Gg of Hg in the upper 1 m. However,
measurements of $R_{HgC}$ on Siberian permafrost peatlands are largely missing, leaving the size of
estimated northern soil Hg budget, and its fate under arctic warming scenarios uncertain. Here
we present Hg and carbon data for 6 peat cores, down to mineral horizons at 1.5 - 4 m depth,
across a 1700 km latitudinal (56 to 67°N) permafrost gradient in the Western Siberian lowlands
(WSL). Hg concentrations increase from south to north in all soil horizons, reflecting enhanced
net accumulation of atmospheric gaseous Hg by the vegetation Hg pump. The $R_{HgC}$ in WSL peat
horizons decreases with depth from 0.38 Gg Pg$^{-1}$ in the active layer to 0.23 Gg Pg$^{-1}$ in
continuously frozen peat of the WSL. We estimate the Hg pool (0-1 m) in the permafrost-
affected part of WSL peatlands to be 9.3 ± 2.7 Gg. We review and estimate pan-arctic organic
and mineral soil $R_{HgC}$ to be 0.19 and 0.77 Gg Pg$^{-1}$, and use a soil carbon budget to revise the
northern soil Hg pool to be 67 Gg (37-88 Gg, interquartile range (IQR)) in the upper 30 cm, 225
Gg (102-320 Gg) in the upper 1 m, and 557 Gg (371-699 Gg) in the upper 3 m. Using the same
$R_{HgC}$ approach, we revise the global  upper 30 cm soil Hg pool to contain 1078 Gg of Hg (842-
1254 Gg, IQR), of which 6% (67 Gg) resides in northern permafrost soils. Additional soil and
river studies must be performed in Eastern and Northern Siberia to lower the uncertainty on
these estimates, and assess the timing of Hg release to atmosphere and rivers.





## 1. Introduction

High-latitude organic-rich soils are key ecosystems controlling the transfer of carbon, nutrients, and pollutants between the atmosphere, rivers, lakes and the Arctic Ocean. These soils are most vulnerable to on-going climate change, due to the high mobility of carbon stored in the form of peat deposits. Part of the peat layers are currently frozen but may be subjected to fast thaw, especially in discontinuous and sporadic permafrost zones (Romanovsky et al., 2010). Whilst the stock of C in Arctic and subarctic peat and mineral soils is fairly well quantified (472 Pg C (±27 Pg, 95% confidence interval, CI) in the upper 0-1m (Hugelius et al. 2014), this is not true for pollutants such as mercury (Hg). Because of its strong bio-amplification in Arctic marine biota (Morel et al., 1998), and exposure to native Arctic populations (AMAP, 2011), there is a strong interest in understanding Hg biogeochemistry in Arctic environments (Outridge et al., 2008; Steffen et al., 2008; Stern et al., 2012).

Recent advances in quantifying Arctic Hg cycling show that Arctic $Hg^{II}$ wet deposition is generally low (Pearson et al., 2019), and that the vegetation Hg pump drives yearlong net gaseous $Hg^0$ (and $CO_2$) deposition, via foliar uptake to Arctic vegetation and litterfall to soils (Obrist et al. 2017; Jiskra et al. 2018; Jiskra et al., 2019). Soil core analyses in Alaska indicate that large amounts of carbon and Hg have accumulated since the last glacial maximum, and two upscaling approaches to Hg stocks in pan-Arctic permafrost soils resulted in differing estimates of 184 Gg and 755 Gg for the upper 1 m (Schuster et al., 2018; Olson et al., 2018). Despite the overall net atmospheric Hg deposition to soils, research has found that Arctic rivers export 44 Mg y$^{-1}$ of soil Hg, bound to particulate and dissolved organic matter, to the Arctic Ocean (Fisher et al., 2012; Dastoor et al., 2014; Zhang et al., 2015; Sonke et al., 2018). Together with coastal erosion of soils (30 Mg y$^{-1}$), river Hg inputs constitute a terrestrial Hg flux of 74 Mg y$^{-1}$ to the Arctic Ocean that is of similar magnitude to gross atmospheric deposition over the Arctic Ocean (80 Mg y$^{-1}$, Sonke et al., 2018). Permafrost thawing has been shown to enhance river Hg export from soils to rivers

(St Pierre et al., 2018), is most pronounced in the discontinuous permafrost zone, and has been
suggested to potentially double over the next 50 years ( Lim et al., 2019). The quantity of
atmospheric Hg deposition to northern peat soils that is presently re-emitted to the atmosphere is
not well understood. Hg exchange studies indicate temporally limited $Hg^0$ emission from the
Alaskan permafrost tundra at 68°N (Obrist et al., 2017), and strong year round net $Hg^0$ emission
from Scandinavian peat at 64°N (Osterwalder et al., 2018). Other studies provide evidence for
vegetation type (Rydberg et al., 2010) and temperature and insolation control (Fahnestock et al.,
2019) on net $Hg^0$ deposition or emission.

All available data of Hg in permafrost soils originate from N-America or Scandinavia

(Jensen et al., 1991; Bailey et al., 2002; Talbot et al., 2017; Schuster et al., 2018; Olson et al.,
2018). Except for two studies of Hg in a peat profile from a permafrost-free zone of western
Siberia (Golovatskaya et al., 2009; Lyapina et al., 2009), we did not find extensive measurements
of Hg in peat profiles from permafrost regions of the Russian Arctic and Siberia. Recent work
used a soil carbon GIS model to estimate the size of the northern permafrost soil Hg inventory to
be 755 ±427 Gg (95% CI)  in the upper 1 m (Schuster et al., 2018). However, this estimate is
based on extrapolation of high Hg to organic carbon (C) ratios, $R_{HgC}$, of 1.6 Gg Hg per Pg of C
(Gg $Pg^{-1}$) in Alaskan mineral soils to the entire N-American and Eurasian permafrost zone. A
second study used lower $R_{HgC}$, of 0.12 to 0.62 Gg $Pg^{-1}$, derived from observations on both Alaskan
organic and mineral soils and literature data, to estimate a lower northern soil 0-1 m Hg inventory
of 184 Gg (136-274 Gg, 37.5-62.5 percentiles) (Olson et al., 2018). Direct measurement of soil
Hg and carbon profiles in frozen peatlands of Siberia are needed to address these variable
estimates, and compare the size of permafrost soil Hg pool to the global soil Hg pool. This
constitutes the first and main objective of the present study. The second objective was assessing
the impact of permafrost type (absent, sporadic, discontinuous and continuous) on Hg
concentrations and pools in the active layer, frozen peat and mineral horizons. The third objective



was to relate Hg concentration in peat to that of other trace metals in order to reveal possible
mechanisms of Hg and other metal pollutant accumulation within the organic and mineral horizons
of frozen peatlands.

**2. Study Site and Methods**
*2.1.Sampling sites*

Soil sampling was performed along a latitudinal transect of the western Siberia lowlands

(WSL) that comprised the southern taiga (Plotnikovo, 56°N), the middle taiga (Mukhrino, 60°N),
the northern part of the taiga zone (Kogalym, 62°N), forest-tundra (Khanymey 62°N and Pangody,
65°N) and tundra (Tazovsky, 67°N) biomes (**Fig. 1**). In the WSL, the permafrost zones follow the
temperature and vegetation distribution over the latitude at otherwise similar relief, lithology and
runoff, thus allowing to test the effect of permafrost by analyzing latitudinal features of Hg
distribution in soils. Key physico-geographical parameters of studied sites and soil types are listed
in **Table S1** of the Supplementary Information. The WSL peat actively formed since the beginning
of the Holocene until freezing of bogs in the sub-Boreal period (11-4.5 ky, Kremenetski et al.,
2003; Panova et al., 2010; Ponomareva et al., 2012; Loiko et al., 2019). Since 4.5 ky, the rate of
peat formation and bog extension in the permafrost-affected part of the WSL have decreased. In
the southern part of cryolithozone and permafrost-free part of WSL, peat accumulation and bog
extension remained active over the entire Holocene (Kurina and Veretennikova, 2015; Preis and
Karpenko, 2015; Kurina et al., 2018). The main mineral substrates underlying frozen peat layers
of the WSL are quaternary clays, sands, and alevrolites. In the southern part (sites Plotnikovo and
Mukhrino), the typical substrate is carbonate-bearing clays of lake-alluvium origin with rare layers
of sandstones (**Table S1**).

Mean annual atmospheric temperature (MAAT) increases from south to north, being equal

to –0.4, –1.2, –4.0, –5.6, –6.4, and –9.1°C at Plotnikovo, Mukhrino, Kogalym, Khanymey,





Pandogy and Tazovsky, respectively (Trofimova and Balybina, 2014). The permafrost is absent
in Plotnikovo but present at all other sites and ranges from relict to isolated (Mukhrino), isolated
to sporadic (Kogalym) in the south, to discontinuous (Khanymey, Pangody) and continuous
(Tazovsky) in the north. At permafrost-affected sites, the average active (unfrozen) layer thickness
(ALT) at the time of sampling of peat mounds (hummocks) ranged from 90 cm in the south to 45
cm in the north. The peat mounds of ombrotrophic bogs probed in this work are present across the
full latitudinal gradient.
The vegetation of three studied types of bogs (polygonal, flat-mound and ridge-hollow) is
essentially oligotrophic (poor in nutrients) which indicates the ombrotrophic (rain and snow water
fed) conditions, i.e., the lack of groundwater input and lateral surface influx of nutrients. The flat-
mound palsa is covered by dwarf shrubs (*Ledum decumbens*, *Betula nana*, *Andromeda polifolia,*
*Vaccinium ssp., Empetrum nigrum)*, lichens (*Cladonia ssp., Cetraria, Ochrolechia)* and mosses
(*Dicranum ssp., Polytrichum ssp., Sphagnum angustifolium, S. lenense*). At southern sites, the
pine *Pinus sylvestris* is abundant on ridges (Peregon et al., 2008, 2009) whereas the two taiga sites
are dominated by *Pinus sylvestris f. uliginosa* with minor but permanently present *Betula*
*pubescens* and *Pinus sibirica*. Dwarf shrubs are dominated by *Ledum palustre, Chamaedaphne*
*calyculata, Vaccinium vitis-idaea*. The moss layer is dominated by *Sphagnum fuscum*, *S.*
*angustifolium* with the presence of *Sphagnum magellanicum, S. capillifolium* and boreal forest
moss species like *Pleurozium schreberi*.

*2.2. Sampling procedure, analyses and data treatment*
Peat core samples were collected in August, when the depth of unfrozen layer was at its
maximum (i.e., see Raudina et al., 2017). Based on measurements by temperature loggers over
the summer period, the in-situ temperature of studied soil profile ranged from 15±5°C in the top
soil (0-20 cm) to 4±2°C at the permafrost boundary (40-80 cm). The physical, chemical and





botanical properties of several peat cores collected in the homogeneous palsa region in the north
and ridgre-ryam complex in the south are highly similar among different peat mounds (Velichko
et al., 2011; Stepanova et al., 2015).

Peat cores were extracted using a Russian sediment/peat corer and the frozen part was

sampled using a motorized Russian peat corer (UKB-12/25 I, Russia) with a 4-cm diameter corer
sterilized with 40% ethanol prior to each extraction. We collected the full active (unfrozen) layer
peat column, the frozen peat column and some 10 to 30 cm of frozen mineral horizons using clean
powder-free vinyl gloves. Peat or mineral soil samples were divided in 5-10 cm segments using a
sharp sterile single-use plastic knife. Soil samples were placed in sterile PVC doubled-zipped bags
and kept at -20°C during transport and storage. To avoid contamination of peat from external
surroundings, we separated the part of the core for geochemical analysis exclusively from the
interior of the core (> 1 cm from the core liner) following conventional procedures (Wilhelm et
al., 2011).

Total Hg concentration, THg, in freeze-dried and ground slices of peat cores was

determined using a direct mercury analyzer (DMA-80 - Milestone, Italy). Analysis of reference
material BCR-482 (lichen, 480±28 ng g$^{-1}$), MESS3 (sediment, 91±9 ng g$^{-1}$) and NIST 1632d (coal,
93±3 ng g$^{-1}$) showed good reproducibility (mean±1σ) of 467±28 ng g$^{-1}$, 80±6 ng g$^{-1}$ and 98±8 ng
g$^{-1}$, respectively. The average uncertainty on duplicate sample analysis did not exceed 5% (1σ).
The carbon (C) and nitrogen (N) concentration were measured using catalytic combustion with
Cu-O at 900°C with an uncertainty of ≤ 0.5% using Thermo Flash 2000 CN Analyzer, and aspartic
acid (C 36.09%±1.5%; N 10.52%±0.5%) and soil SRM (C 2.29%±0.07%; N 0.21%±0.01%) as
standards. Analyses of total C before and after sample treatment with HCl did not yield more than
1 % of inorganic C; therefore our total C-determination represents organic carbon. For trace and
major element analysis, soil samples were subjected to full acid digestion in the clean room
following ICP-MS (Agilent 7500 ce) analyses as described previously (Morgalev et al., 2017).



The Shapiro Wilk normality test was used to asses THg, elemental and $R_{HgC}$ distributions,
and statistical data descriptors adjusted accordingly. All statistical tests used a significance level
of 95% ($\alpha = 0.05$). Spearman rank order correlations (significant at $p < 0.05$) were performed to
characterize the link of Hg with C, N and other major and trace elements. The differences in Hg
concentration between the active- and frozen peat layer were tested using the Mann-Whitney U
test for paired data at a significance level of 0.05.
C pools of different soil classes reported by Hugelius et al. (2014) were divided into two
categories, organic and mineral soils. Histosols and Histels were defined as organic soils. Turbels
and Orthels were considered as organic soils for the $0 – 0.3$ m interval and as mineral soils for the
$0.3 – 3$m interval. All other soils were considered as mineral soils. To estimate the northern soil
Hg pool, C pools were multiplied with the respective $R_{HgC}$ derived for organic (>20% C) and
mineral (<20% C) soil data from north America (excluding Alaska) and Eurasia. To calculate the
global Hg pool, a simpler approach was used and one singe $R_{HgC}$ was considered for 5 climate
zones which were defined by latitude (arctic: $> 67°$, boreal: $50° - 67°$, temperate: $35° - 50°$,
subtropical: $23.45° - 35°$, tropical: $<23.45°$) according to FAO and ITPS (2018). The uncertainty
was assessed with a Monte carlo approach using the *rnorm* and *rlnorm* function of R (version
3.6.1.) and is reported as the interquartile range (25th and 75th percentile) of 100,000 simulations.
For the northern soil Hg pool, final uncertainties incorporate the uncertainties on the C stock from
Hugelius et al. (2014) assuming normal distribution and uncertainties of $R_{HgC}$ assuming log-
normal distribution.







### 3. Results

*3.1. Depth (vertical) distribution of Hg in peat profiles*

Hg concentration in peat cores of the WSL are illustrated in **Fig. 2** and primary data on soil chemical composition and Hg concentration are listed in **Table S2** of the Supplement. The upper 0-20 cm layer is 2 to 3 times enriched in Hg compared to the rest of the peat core in permafrost-affected sites (Khanymey, Pangody and Tazovsky). This is not the case, however, for the sporadic permafrost zone (Kogalym) and isolated permafrost zone (Mukhrino), where a local maximum at ca. 35 cm depth was detected but no enrichment of upper 10-20 cm horizons occurred. In the most southern, permafrost-free site of the WSL (Plotnikovo, southern taiga), the Hg concentration profile in the peat was fairly constant with a local minimum at 100 cm depth. The mean, depth-integrated Hg concentrations in active layer, permafrost and mineral horizons are illustrated in **Fig. 3** and summarized in **Table 1**. The latitudinal trend of Hg concentration in peat consists of a systematic increase northward, both in permafrost and active peat layers. The dominant ground vegetation (lichens) analyzed at 5 sites out of 6 (Plotnikovo, Kogalym, Khanymey, Pangody and Tazovsky) did not show significantly different (U test Mann-Whitney) Hg concentrations relative to the peat cores (**Fig. 3**). The typical concentrations of Hg in studied peat cores ranged from 7 to 284 ng $g^{-1}$ with a median (± IQR) of 67±57 ng $g^{-1}$. The Hg concentration in the thawed, active layer was generally comparable to that in the frozen layer, supported by a Mann-Whitney test, which did not show significant difference in Hg concentration between frozen and thawed peat in all permafrost-affected sites. Within the latitudinal transect from south to north, the Hg concentrations in peat are higher (Plotnikovo, Kogalym, Khanymey, Pangody) or comparable (Tazovsky) to those in the mineral horizons.

The ratio of Hg:C ($R_{HgC}$, µg $g^{-1}$, corresponding to Gg $Pg^{-1}$) ranged between 0.05 and 2.0 over the peat columns, and was 5 to 10 times higher in mineral horizons compared to frozen peat and active layers (**Fig. 4**). The $R_{HgC}$ in the active layer and in the mineral horizons increased 3-



fold from the south (56°N) to the north (67°N). In the frozen peat horizon, the $R_{HgC}$ ratio increased
two-fold from sporadic and isolated to continuous permafrost zone.

*3.2. Regional and total pools of Hg in the WSL peat and mineral layers*

The mass of Hg per area of soil in the active- and frozen peat layer as well as in the top 30

cm of frozen mineral horizons of the six studied WSL peat profiles was calculated by multiplying
bulk soil peat and mineral layer densities (range from 0.01 to 0.38 g/cm$^3$, **Table S2**) by Hg
concentration and integrating over the corresponding depths. The surface area - normalized Hg
stock systematically increased from south to north (ca. 0.3 to 6.0 mg Hg m$^{-2}$ and ca. 0.8 to 13.7
mg Hg m$^{-2}$, in the 0-30 and the 0-100 cm peat layer, respectively (**Fig. 5 A**). This northward
increase was most pronounced for the active layer, was less evident for frozen peat, and
insignificant for the upper 30 cm of mineral horizon located under the peat (**Fig. 5 B**). Taking into
account the proportion of bogs (peatlands)  in each zone (1° latitudinal grid) from Sheng et al.
(2004), we calculated the pool of Hg in permafrost-free and permafrost-affected WSL peatlands
(**Fig. 6**). The total pool of Hg in the 0-100 cm layer of peat bogs exhibits a maximum (356-580
Mg) in the discontinuous permafrost zone.

We estimate the total organic soil Hg pool in the WSL from the Hg stock (mg Hg m$^{-2}$ over

0-100 cm depth) for permafrost and permafrost-free zones (**Fig. 7 A**), extrapolated to the full
average thickness of peat in the WSL (280 cm, Sheng et al., 2004), assuming that Hg concentration
in the upper 0-100 cm peat layer is the same as in 100-280 cm of peat and multiplied by the area
of bogs in each latitudinal grid (S, m²) as shown in **Fig 7 B**. This yields 1.7 Gg Hg in the
permafrost-free zone and 7.6 Gg Hg in the permafrost-bearing zone with a total Hg pool of 9.3
Gg in the WSL. For this calculation we did not take into account the mineral horizons and we used
variable active layer thickness across the latitudinal gradient of WSL, as estimated at our sampling
sites (**Table S1**). The amount of Hg in permafrost-bearing zone  within the active (unfrozen) peat





layer (0-160 cm in the south and 0-20 cm in the north) of the WSL is 2.0 Gg, and that in the frozen
(160-280 cm in the south and 20-280 cm in the north) layer) is 5.6 Gg.

Alternatively, to calculate the total pool of Hg in WSL bogs, we used the $R_{HgC}$ inferred

from our data across the gradient of permafrost and biomes (Table 1). Taking into account the C
pool in the WSL (70.2 Pg C of 0-280 cm depth layer, Sheng et al., 2004) and the median $R_{HgC}$ of
0.133 µg/g in the WSL, we calculated Hg for the full depth of the peat layer in each zone. This
also gives 9.3 Gg Hg for a total area of 592,440 km².

*3.3. Correlation of Hg with other elements in the peat cores*

Spearman rank order correlations of Hg with other elements demonstrated significant

positive relationships ($R > 0.60$; $p < 0.05$) with K, Rb, Cs, P, As, W, V, Cr, Cu in the active
(unfrozen) layer (**Table S3** of Supplment). However, these relationships were less pronounced in
the frozen peat, where only Mg, Ca, Sr, Mn, N, P, As, Cu, Ni, Sb and some REE demonstrated
minor ($0.40 < R < 0.55$) positive correlations with Hg. Finally, in the mineral layer, significant
($R > 0.70$) positive correlations of Hg were observed with Li, Ca, Sr, P, N, Mn, Ni, Co, Cr, Cd. A
positive ($R = 0.60$) relationship between Hg and C was observed in mineral horizons, whereas no
correlation was detected in both frozen and thawed peat. This is consistent with some studies of
peat soil in Brazil (Roulet et al., 1998) and Arctic tundra soils (Olson et al., 2018). At the same
time, there was a positive correlation of Hg with N in the active layer, frozen peat and mineral
horizons ( $R = 0.50, 0.47$ and $0.75$, respectively). Stronger and more stable correlation of Hg with
N compared to C was also noted by Roulet (2000).








## 4. Discussion

### *4.1. Hg association with other elements in peat*

Stronger accumulation of Hg relative to C in mineral horizons in the north (Tazovsky, **Fig. 4**) may be linked to the clay nature of mineral layers (Roulet et al., 1998; Baptista-Salazar et al., 2017) in these regions (**Table S1**) but also to the presence of specific host phases of Hg (see examples of peat minerals in Rudmin et al., 2018). Dissolved oxygen measurements in soil porewaters at the Tazovsky site indicate that mineral gleysoils and peat histosols, which often overlay former lake sediments, are anoxic (Raudina et al., 2017; Loiko et al., 2019). The Hg host phases in these soils are therefore likely sulfide minerals. Indeed, known Hg carriers in peat deposits are Fe and Zn sulfide minerals or organic-bound sulfide functional groups (Smieja-Król et al., 2010; 2014; Prietzel et al., 2009; Skyllberg et al., 2003, Bates et al., 1998; Steinmann and Shotyk, 1997).

In the peat active layer, Hg was positively correlated with K, Rb, Cs, P, As, V, Cr, Cu (**Table S3**). In the frozen part of the peat core, Hg was positively correlated with Ca, N, Mn, Sr, Mg, P (**Table S3**). Indeed, atmospheric particles in snow across the WSL exhibit strong enrichment in Mo, W, As, Sb, Ni, Cu, Zn, Cd, Pb, Mg, Ca, and Na (Shevchenko et al., 2017). The strong positive correlation of Hg with these elements in peat soils of WSL suggests a common atmospheric origin. Note however, that the cited elements deposit with particles, rainfall and snowfall, whereas atmospheric Hg transfer to peat occurs mainly via the vegetation pump, with tundra and taiga vegetation actively taking up atmospheric gaseous $Hg^0$ through foliage (Obrist et al., 2017; Jiskra et al., 2018).

### *4.2. Estimating the northern soil Hg pool*

A recent study used a median $R_{HgC}$ value of 1.6 µg g$^{-1}$, observed mainly in mineral soil samples (median SOC of 3%, IQR=1.7 to 8.7 %) along a transect in Alaska, to estimate a northern





permafrost soil Hg pool of 755±427 Gg in the upper 0-100 cm, and 1656±962 Gg in the upper 0-
300 cm (Schuster et al. 2018). In the case of western Siberia, this high $R_{HgC}$ value overestimates
the Hg pool 12-fold, given that the median $R_{HgC}$ in WSL peat is only 0.13±0.12 (median±IQR)
(**Table 1, Fig. 4**). The extrapolation based on Alaskan $R_{HgC}$ for the whole Northern Hemisphere
permafrost region also suggests that the WSL contains large amounts of Hg in the upper 0-30 cm
(20-40 mg Hg m$^{-2}$) and in the upper 0-100 cm (40-80 mg Hg m$^{-2}$). These numbers are much higher
than the direct measurements in this study: 0.3 mg Hg m$^{-2}$ in 0-30 cm and 0.8-1.3 mg Hg m$^{-2}$ in
0-100 cm layer in the permafrost-free zone (Plotnikovo and Mukhrino sites); 0.5 mg Hg m$^{-2}$ in 0-
30 cm and 3.0 mg Hg m$^{-2}$ in 0-100 cm layer in the sporadic zone (Kogalym site); 1.8-4.0 mg Hg
m$^{-2}$ in 0-30 cm and 9.6-11.9 mg Hg m$^{-2}$ in 0-100 cm layer in the continuous to discontinuous
permafrost zone (Khanymey and Pangody sites), and 6.0 mg Hg m$^{-2}$ in 0-30 cm and 13.7 mg Hg
m$^{-2}$ in 0-100 cm layer in continuous permafrost zone (Tazovsky). It is worth noting that the recent
data of Talbot et al. (2017) for Ontario (Canada) bogs ($S_{area}$ = 1,133,990 km²; 18.8 Gg Hg for the
277 ± 123 cm depth) are consistent with the results of the present study in the WSL (9.3 Gg Hg
for 592,440 km² for the 280 ± 100 cm depth).

318         A revised value of the Hg pool in the northern soils was recently provided by Olson et al.

(2018) who combined measured $R_{HgC}$ values for Alaskan tundra soils with literature data, and
derived $R_{HgC}$ of 0.12 µg g$^{-1}$ for 0-30 cm (organic) and 0.62 µg g$^{-1}$ for 30-100 cm (mineral) layers.
Olson et al. (2018) estimate northern permafrost soil Hg pools of 26 Gg (0-30 cm) and 158 Gg
(30-100 cm), which combined (184 Gg) is 4 times lower than the number of 755 Gg (0-100 cm)
by Schuster et al. (2014). Both studies rely on $R_{HgC}$ measurements from Alaskan soils, due to
relatively easy road access to the sampling sites along the Dalton Highway. Bedrock along the
Dalton Highway contains relatively high geogenic Hg levels (mean concentration: 32 ng/g),
resulting in a high geogenic contribution in mineral soils (39% for B horizons and 20% for A
horizons, Obrist et al., 2017). As a result, $R_{HgC}$ in mineral soils along the Dalton highway are



higher (median = 1.6, Schuster et al. 2018) than for other mineral soils in North Amercia and
Eurasia (median = 0.64, **Figure 8**). It is clear that any upscaling calculation of pan-Arctic
permafrost Hg depends critically on the $R_{HgC}$ of the 0-30 and 30-100 cm peat layers, as Eurasian
sporadic to continuous permafrost represents 54% of the northern soil C inventory (**Table 2**).
Compared to the previously assumed $R_{HgC}$ of 1.6 (Schuster et al. 2018), and 0.12 - 0.62 µg g$^{-1}$
(Olson et al.., 2018), we observe lower $R_{HgC}$ ranging from 0.065 to 0.38 µg g$^{-1}$ at 0-100 cm depth.
Setting Alaska aside as a geographic region, we find that North American and Eurasian mineral
(<20% SOC) soil $R_{HgC}$ was lower (µ = 0.77 µg g$^{-1}$, median= 0.63 µg g$^{-1}$ (IQR = 0.32 to 0.80 µg g$^{-1}$
$^{1}$), n=131) than $R_{HgC}$ reported for Alaska (median= 1.64 µg g$^{-1}$ (IQR = 0.91 to 2.93 µg g$^{-1}$), n=589)
(**Figure 8**). The $R_{HgC}$ in organic soils (>20% SOC, including data from Alaska) was approximately
4 times lower (µ= 0.19 µg g$^{-1}$, median= 0.15 µg g$^{-1}$ (IQR = 0.09 to 0.24 µg g$^{-1}$), n=449) than that
in mineral soils of North America and Eurasia (**Figure 8**), consistent with the observed difference
in WSL mineral and organic soils. Higher $R_{HgC}$ observed in mineral soils may originate from a
contribution of geogenic Hg from the weathered bedrock (independent of C stock) and/or a higher
mineralization rate of C (preferential C over Hg loss) in predominantly oxic mineral soils
compared to anoxic peat soils.

In **Table 2** we revisit the full 0-300 cm northern permafrost soil Hg inventory, based on

N-American (excluding Alaska) and Eurasian $R_{HgC}$ based on the literature data compilations of
Olson et al. (2018) and Schuster et al. (2018), and our observed WSL $R_{HgC}$ for Eurasia, multiplied
by estimated northern tundra soil organic C pools for 0-300 cm from and Hugelius et al. (2014).
The error made by neglecting high $R_{HgC}$ in Alaskan mineral soils is small, on the order of 2.5 Gg
Hg, as estimated from the relatively small Alaskan C pool of 2.6 Pg C (Tarnocai et al., 2009). We
estimate the northern soil Hg pool to be 67 Gg (37-88 Gg, IQR) in the upper 30 cm, 225 Gg (102-
320 Gg) in the upper 1 m, and 557 Gg (371-699 Gg) in the upper 3 m (**Table 3**). Note that our
revised values in the 0 - 1m range (225 Gg) is similar to that of Olson et al. (184 Gg), but lower





than that of Schuster et al. (755 Gg). We find that Hg stocks in organic soils (>20% SOC) represent
56% and 21% of the total Hg stock in the 0-30 cm and 0-100 cm depth range, respectively (**Table**
**2**). The rest of the pan-arctic Hg is associated with C in mineral soils (<20% SOC) for which
relatively sparse data exists (n=131). In particular, turbel and orthel mineral soils, which are
estimated to contain 49 to 62% of total arctic C (Hugelius et al., 2014) and 36 to 85% of Hg at the
various depth intervals need to be further investigated.

*4.3. Estimating the global soil (0-30 cm) Hg pool*
To estimate the global soil Hg pool, we combined the more detailed Arctic pool estimate
(separating organic and mineral soils) with a more basic approach for the other climate zones,
where we derived bulk $R_{HgC}$ for the 0-30 cm surface soils based on published literature data and
multiplied it with global C stock estimates for each climate zone (arctic, boreal, temperate,
subtropical and tropical) from the global soil organic carbon map (FAO and ITPS, 2018). The
$R_{HgC}$ increases from cold climate zones to warmer climate, from 0.15 µg g$^{-1}$ for Arctic organic
soils, to 1.8 µg g$^{-1}$ in subtropical and tropical soils (**Figure 9, Table 4**). This latitudinal trend in
$R_{HgC}$  likely reflects a combination of low C mineralization rates in colder north and additional Hg
sorption to Fe(oxy)hidroxides in old tropical soils. Taking into account the variation in $R_{HgC}$ and
C stocks across the climate zones, we estimate a global Hg stock of 1084 Gg (848 – 1258 Gg,
IQR) for the top 0 - 30 cm (**Table 4**). Previous global Hg soil pool estimates vary between 232
and 1150 Gg (Selin et al., 2008; Smith-Downey et al., 2010; Amos et al., 2013, 2015; Hararuk et
al., 2013; Wang et al., 2019). Schuster et al. (2018) concluded that Arctic permafrost soils store
nearly twice the amount of Hg as all other soils, the Ocean and atmosphere combined, but in doing
so they compared different global (0 - 30cm) and Arctic soil depth ranges (0 - 300cm). Our revised
estimate of the pan-Arctic permafrost and global soil pool suggests that, for a similar depth range
of 0-30 cm, permafrost soils contain 6% (67 Gg) of the global soil Hg pool (1084 Gg).



*4.4. Northern soil Hg sequestration and Hg loss*

Olson et al. (2018) recognized that the large 0-100 cm northern soil Hg pool is the result of thousands of years of net atmospheric Hg deposition. The latitudinal trend of northward increasing peat Hg concentration in the WSL (**Fig. 4, 5**) illustrates that this net Hg deposition is a fine balance between the vegetation Hg pump, which sequesters $Hg^0$ in soils via foliar uptake and litterfall, and $Hg^0$ emission during biomass decay of vegetation debris. Annual gross Hg sequestration by vegetation, via the vegetation pump, likely scales with primary productivity and therefore decreases northward as insolation and growing season decrease. However, in the north, degradation rates of vegetation biomass are lower than in the south: the moss biomass losses during decomposition in the forest tundra zone (5-6% over 1st year and 10-12% over 2 years) is lower than that in the southern taiga (10-20% over 1st year and 20-40% over 2 years), based on in-situ biomass degradation experiments across the WSL gradient of biomes (Vishnyakova and Mironycheva-Tokareva, 2018). The net result is a higher preservation of soil Hg in the north, where less emission of $Hg^0$ during plant decay occurs. The dependence of this balance, between $Hg^0$ sequestration and $Hg^0$ re-emission, on climate explains qualitatively the contrasting observations made in Toolik (AK, USA, 68°N, MAAT = -7°C, Obrist et al., 2017) and Degerö Stormyr (Sweden, 64°N, MAAT = 2°C, Osterwalder et al., 2018). At Toolik, net $Hg^0$ deposition by vegetation and soil uptake occurs on an annual basis, whereas at Degerö Stormyr higher temperatures result in net annual $Hg^0$ emission. More research is needed to quantify the climate dependence of $Hg^0$ sequestration (as soil $Hg^{II}$) and $Hg^0$ re-emission before we can predict and model northern soil Hg loss to the atmosphere due to global warming trajectories.



**5. Conclusions**
Western Siberian peatlands contain a large amount of Hg in frozen and thawed peat; the
lateral pools of peat palsa bogs range from 1-2 mg Hg m$^{-2}$ in the south to 10-15 mg Hg m$^{-2}$ in the
north. This northward increase of Hg concentration and pools can be explained by better
preservation of organic-bound Hg due to colder temperatures and shorter active period in the
continuous permafrost zone compared to the discontinuous and sporadic zones. We revisited the
full 0-300 cm northern permafrost soil Hg inventory, based on published $R_{HgC}$ and our observed
WSL $R_{HgC}$ for Eurasia, together with estimated northern tundra soil organic C pools for 0-300 cm
from Hugelius et al. (2014). We estimate the 0-300 cm northern permafrost soil Hg inventory to
be 557 Gg (371-699 Gg, IQR), which is three times lower than a previous estimate of $1656 \pm 962$
Gg Hg for the same depth range (Schuster et al., 2018). We estimate the global soil Hg pool to be
1084 Gg for the 0-30cm depth range. The permafrost Hg pool for the same 0 - 30cm depth range
is 67 Gg, and while large compared to the 3 Gg of Hg residing in the Arctic Ocean (Soerensen et
al., 2016), it represents only 6% of the global soil Hg pool.

**Data availability.** Hg and C concentration data of the WSL soil samples are available in the
supplement. The permafrost data from Schuster et al. 2018 and a global compilation of $R_{HgC}$ data
is available as supplementary information
(https://agupubs.onlinelibrary.wiley.com/doi/full/10.1002/2017GL075571, last access: 6
December 2019). The data from the Olson et al. 2018 study is available from the corresponding
author upon request. The data from the tropical climate zone was compiled from original
publications of Almeida (2005); Almeida et al. (2005); Campbell et al. (2003); Melendez-Perez
et al. (2014).



**Author contributions.** OSP and SVL designed the study. SVL, AGL, and NK performed field
sampling. AGL and OSP did all laboratory analysis. MJ and JES did the northern soil and global
soil Hg pool calculations. All authors contributed to writing of the manuscript.

**We declare no competing interests**

**Acknowledgements:**
This work was mainly supported by the state task of the Ministry of Science and Higher Education
of the Russian Federation, grant no. 1.8195.2017/P220. Partial support from the Russian Fund for
Basic Research, via grant no. 19-29-05209-mk and 19-55-15002-NCNI_a, and from the CNRS
Chantier Arctique Français, via the PARCS project, and the H2020 ERA-PLANET (689443)
iGOSP and iCUPE programmes are acknowledged. M.J. acknowledges funding by the Swiss
National Science Foundation (grant no. PZ00P2_174101). We are thankful to Luiz D. Lacerda for
sharing Hg and C data from Brazil.

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





peatbogs.

| Horizons | C, % | Hg, ng g$^{-1}$ | kg C m$^{-2}$ | mg Hg m$^{-2}$ | $R_{HgC}$ (µg g$^{-1}$) |
|---|---|---|---|---|---|
| Plotnikovo (Pl), Southern taiga, 56.9°N* | | | | | |
| ALT (0-140 cm) | 45±2 | 36±12 | 24 | 2.8 | 0.08±0.03 |
| Mineral (140-150 cm) | 13 | 53 | 23 | 9.0 | 0.40 |
| Total (150 cm) | 43±8 | 37±12 | 47 | 11.8 | 0.10±0.09 |
| 0-30 cm | 44±2 | 44±5 | 3 | 0.3 | 0.10±0.01 |
| 0-100 cm | 45±2 | 36±10 | 9 | 0.8 | 0.08±0.02 |
| Mukhrino (Mh), Middle taiga, 60.9°N* | | | | | |
| ALT (0-360 cm) | 53±7 | 26±13 | 67 | 4.3 | 0.05±0.03 |
| Mineral (360-380 cm) | 15±19 | 24±19 | 51 | 8.3 | 0.46±0.46 |
| Total (380 cm) | 51±10 | 26±19 | 118 | 12.6 | 0.07±0.12 |
| 0-30 cm | 50±0.4 | 29±7 | 4 | 0.3 | 0.06±0.02 |
| 0-100 cm | 52±5 | 32±17 | 18 | 1.3 | 0.06±0.04 |
| Kogalym (Kg), Northern taiga, 62.3°N* | | | | | |
| ALT (0-175 cm) | 48±4 | 48±30 | 93 | 8.7 | 0.10±0.06 |
| Mineral (175-190 cm) | 10±13 | 12±10 | 17 | 2.3 | 0.34±0.36 |
| Total (190 cm) | 45±12 | 45±30 | 110 | 11 | 0.12±0.12 |
| 0-30 cm | 45±1 | 65±19 | 3 | 0.5 | 0.14±0.04 |
| 0-100 cm | 47±2 | 49±34 | 38 | 3 | 0.11±0.07 |
| Khanymey (Kh), Northern taiga, 63.8°N | | | | | |
| ALT (0-34 cm) | 44±2 | 64±43 | 17 | 2.1 | 0.15±0.10 |
| PF1 (34-100 cm) | 50±2 | 47±13 | 78 | 7.6 | 0.09±0.02 |
| PF2 (34-138 cm) | 48±6 | 47±11 | 119 | 11.8 | 0.10±0.02 |
| Mineral (138-147 cm) | 1±1 | 4±1 | 2 | 0.5 | 0.31±0.13 |
| Total (147 cm) | 42±16 | 47±28 | 138 | 14.4 | 0.13±0.09 |
| 0-30 cm | 43±1 | 71±46 | 13 | 1.8 | 0.17±0.11 |
| 0-100 cm | 47±4 | 54±29 | 95 | 9.6 | 0.12±0.07 |
| Pangody (Pg), Forest tundra, 65.9°N | | | | | |
| ALT (0-40 cm) | 50±4 | 78±25 | 38 | 5.3 | 0.16±0.07 |
| PF1 (40-100 cm) | 53±4 | 61 ±25 | 54 | 6.6 | 0.11±0.04 |
| PF2 (40-155 cm) | 48±10 | 67±23 | 78 | 11.0 | 0.15±0.07 |
| Mineral (155-185 cm) | 3±1 | 24±11 | 15 | 12.5 | 0.88±0.28 |
| Total (185 cm) | 41±19 | 62±28 | 130 | 28.8 | 0.27±0.30 |
| 0-30 cm | 50±5 | 83±26 | 26 | 4.0 | 0.17±0.07 |
| 0-100 cm | 52±4 | 68±25 | 92 | 11.9 | 0.13±0.06 |
| Tazovsky (Tz), Southern tundra, 67.4°N | | | | | |
| ALT (0-40 cm) | 49±3 | 186±110 | 22 | 7.4 | 0.38±0.20 |
| PF1 (40-100 cm) | 46±3 | 109±28 | 27 | 6.3 | 0.23±0.06 |
| PF2 (40-380 cm) | 47±4 | 104±39 | 156 | 35.0 | 0.22±0.08 |
| Mineral (380-405 cm) | 14±5 | 152±65 | 60 | 64.7 | 1.24±0.66 |
| Total (405 cm) | 45±9 | 115±57 | 238 | 107.0 | 0.30±0.31 |
| 0-30 cm | 49±4 | 209±120 | 16 | 6.0 | 0.42±0.21 |
| 0-100 cm | 47±3 | 140±80 | 48 | 13.7 | 0.29±0.15 |


**Footnote**: ALT is Active Layer Thickness; PF1 is frozen peat, (ALT-100 cm); PF2 is frozen peat
(ALT to mineral layer); 'Mineral' is mineral layer; 'Total' is total Hg content averaged over full
sampled depth. *In  permafrost-free zone, the ALT extends from the surface to the mineral layer.





**Table 2.** Estimated northern permafrost soil Hg inventory (Gg) for different depth ranges down to 300 cm. Hg pool uncertainties are reported as the interquartile range (IQR), i.e. the $25^{th}$ to $75^{th}$ percentiles of the Hg pool distribution estimates by a Monte Carlo method. Soil organic carbon (SOC) pools are from Hugelius et al. (2014).

| depth range | soils | SOC | Hg Pool | IQR | | Hg % of total per depth range |
|---|---|---|---|---|---|---|
| | | Pg | Gg | Gg | | |
| 0-30 cm | organic (>20% SOC) | 172 | 32 | 12 | 42 | 48 |
| | mineral (<20% SOC) | 45 | 35 | 16 | 45 | 52 |
| | **total** | **217** | **67** | **37** | **88** | |
| 0-100 cm | organic (>20% SOC) | 253 | 47 | 24 | 61 | 21 |
| | mineral (<20% SOC) | 219 | 178 | 58 | 271 | 79 |
| | **total** | **472** | **225** | **102** | **320** | |
| 0-200 cm | organic (>20% SOC) | 366 | 68 | 42 | 86 | 16 |
| | mineral (<20% SOC) | 461 | 364 | 192 | 493 | 84 |
| | **total** | **827** | **433** | **257** | **564** | |
| 0-300 cm | organic (>20% SOC) | 427 | 79 | 52 | 99 | 14 |
| | mineral (<20% SOC) | 607 | 477 | 292 | 621 | 86 |
| | **total** | **1035** | **557** | **371** | **699** | |

**Table 3.** Comparison of estimated northern soil Hg pools by different studies.

| | | | Schuster et al. 2018 | | Olson et al. 2018 | | this study | |
|---|---|---|---|---|---|---|---|---|
| depth range | SOC | 95% CI | Hg Pool | 95% CI | Hg Pool | 25% CI | Hg Pool | IQR[1] |
| | Pg | Pg | Gg | Gg | Gg | Gg | Gg | Gg |
| 0-30 cm | 217 | 12 | 347 | 196 | 26 | 21-42[1] | 67 | 37-88 |
| 0-100 cm | 472 | 27 | 755 | 427 | 184 | 115-232[1] | 225 | 102-320 |
| 0-200 cm | 827 | 108 | 1323 | 764 | | | 433 | 257-564 |
| 0-300 cm | 1035 | 150 | 1656 | 962 | | | 557 | 371-699 |

[1] Confidence interval (CI) corresponding to the 37.5th to 62.5th percentile

**Table 4.** Estimated Hg pool for different climate zones based on reported $R_{HgC}$ and carbon pools. Hg pool uncertainties are reported as the interquartile range (IQR), i.e. the $25^{th}$ to $75^{th}$ percentiles of the Hg pool distribution estimates by a Monte Carlo method.

| Climate zone | C pool | median RHgC | mean RHgC | Hg pool | fraction | IQR | |
|---|---|---|---|---|---|---|---|
| | Pg | (µg/g) | (µg/g) | Gg | (%) | Gg | |
| Tropics[1] | 208 | 1.85 | 2.14 | 446 | 41.2 | 268 | 556 |
| Subtropics[1] | 102 | 1.83 | 2.13 | 217 | 20.0 | 128 | 271 |
| Temperate[1] | 191 | 1.35 | 1.55 | 297 | 27.4 | 186 | 367 |
| Boreal[1] | 140 | 0.36 | 0.42 | 57 | 5.2 | 38 | 69 |
| Arctic[2] | 217[3] | (0.15, 0.64) | (0.19, 0.77) | 67 | 6.2 | 37 | 88 |
| **total** | **858** | | | **1085** | | **848** | **1258** |

[1] Carbon pools are from FAO and ITPS (2018).
[2] The arctic $R_{HgC}$ and Hg pool are from Table 2.
[3] The arctic carbon pool is from Hugelius et al. (2014)





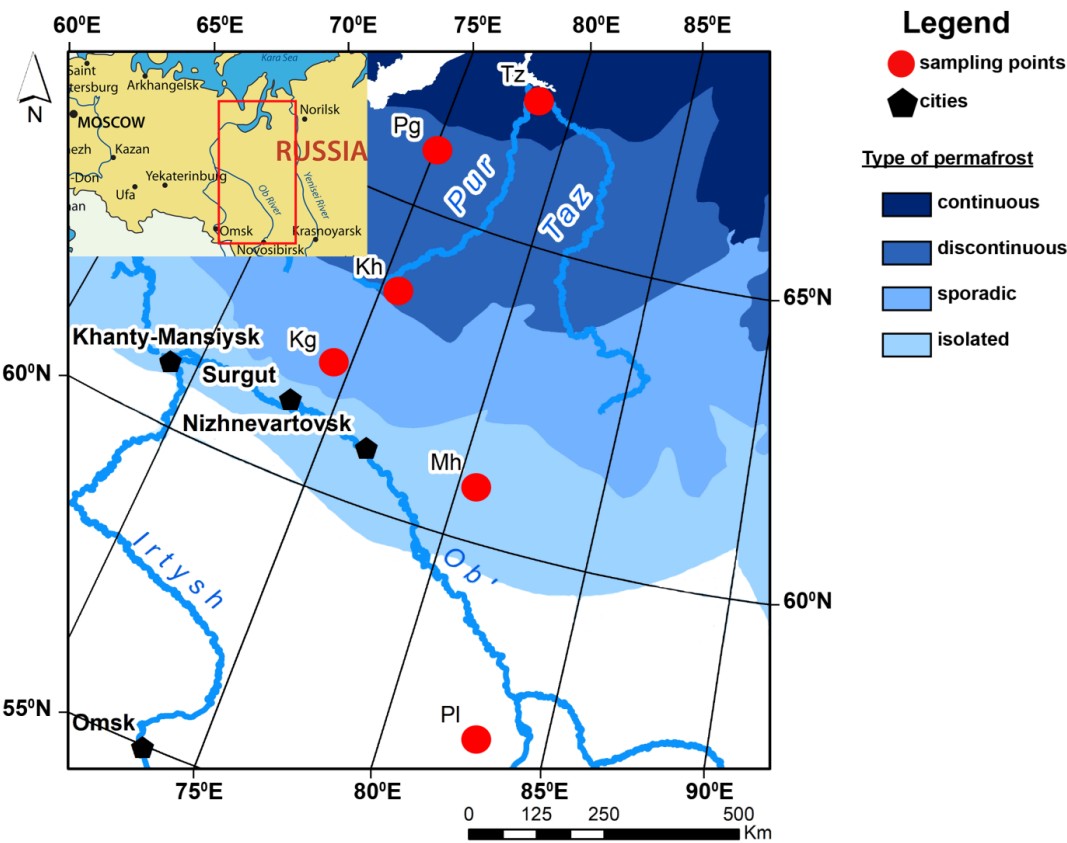

**Fig. 1.** Sampling sites and permafrost boundaries (modified after Brown et al., 2001) of WSL

territory investigated in this work. The climate and soil parameters of 6 sampling sites

(Tazovsky Tz, Pangody Pg, Khanymey Kh, Kogalym Kg, Mukhrino Mh, and Plotnikovo Pl) are

listed in Supplementary Table S1.



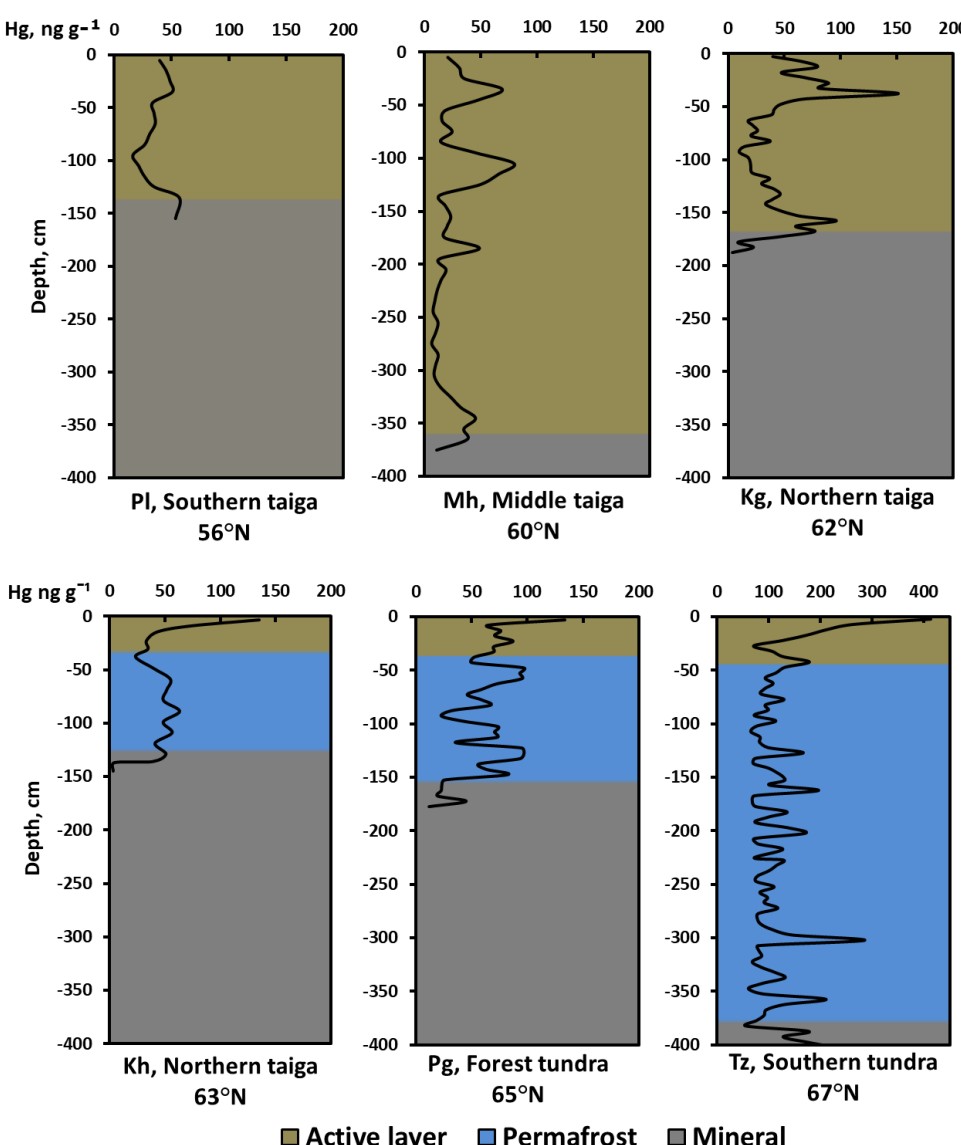

**Fig. 2.** Vertical depth profile distribution of total Hg (THg) in 6 peat cores across a 1700 km latitudinal transect of the WSL. Site location and physio-geographical parameters are shown in Fig. 1 and Supplementary Table S1.


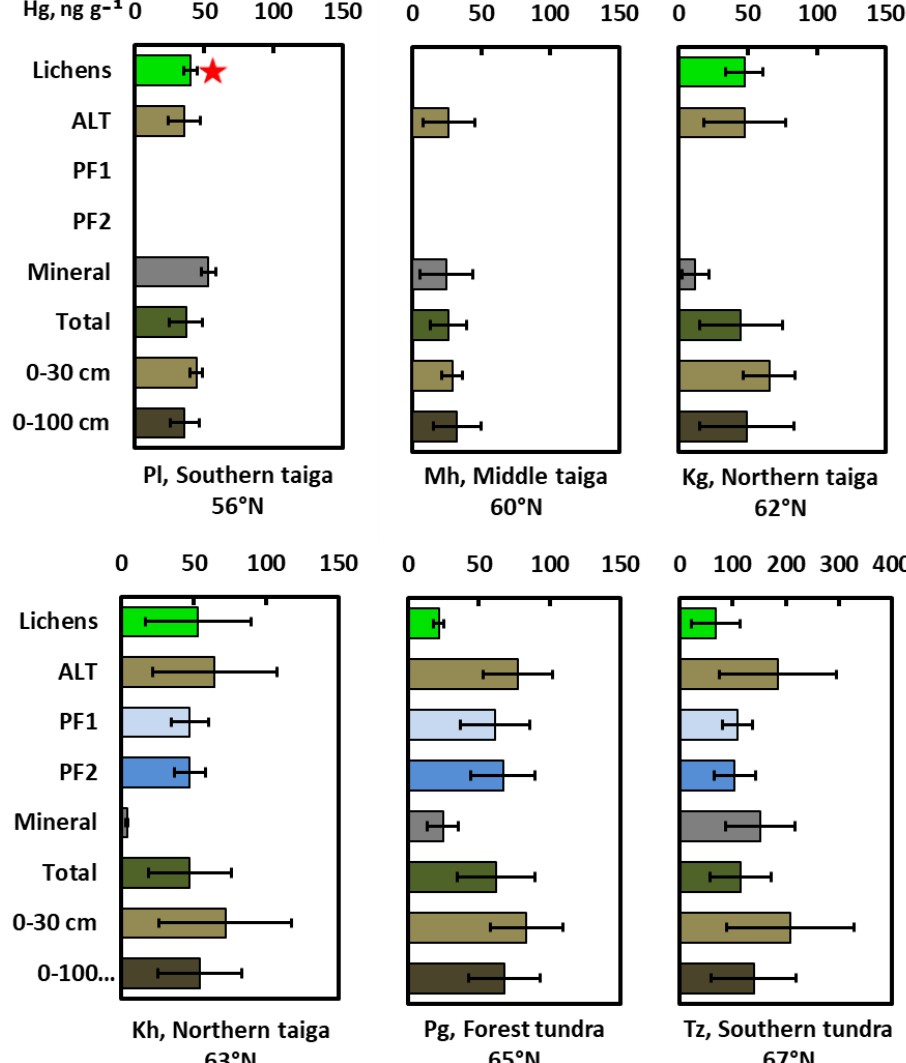


**Fig. 3.** Mean (±SD), depth-integrated Hg concentrations in peat columns and mineral layers of 6
studied sites. Red asterisk represents the data from Lyapina et al. (2009).









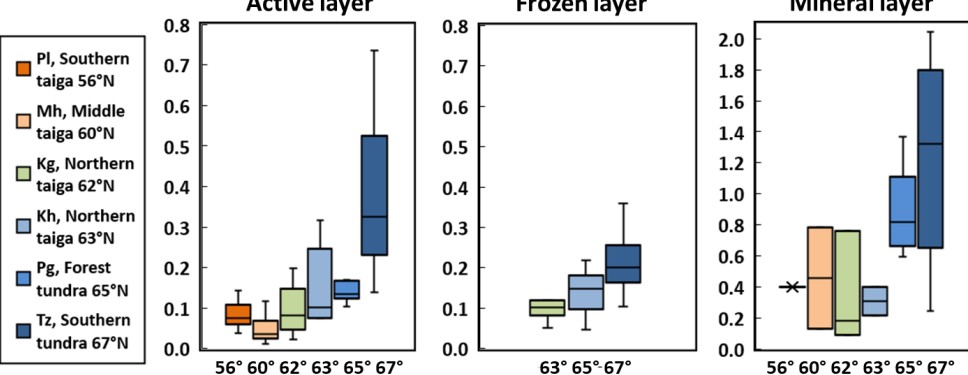



**Fig. 4.** The ratio Hg:C (µg:g), median±IQR, in the active layer, frozen peat and mineral horizons
across the WSL latitudinal transect.
















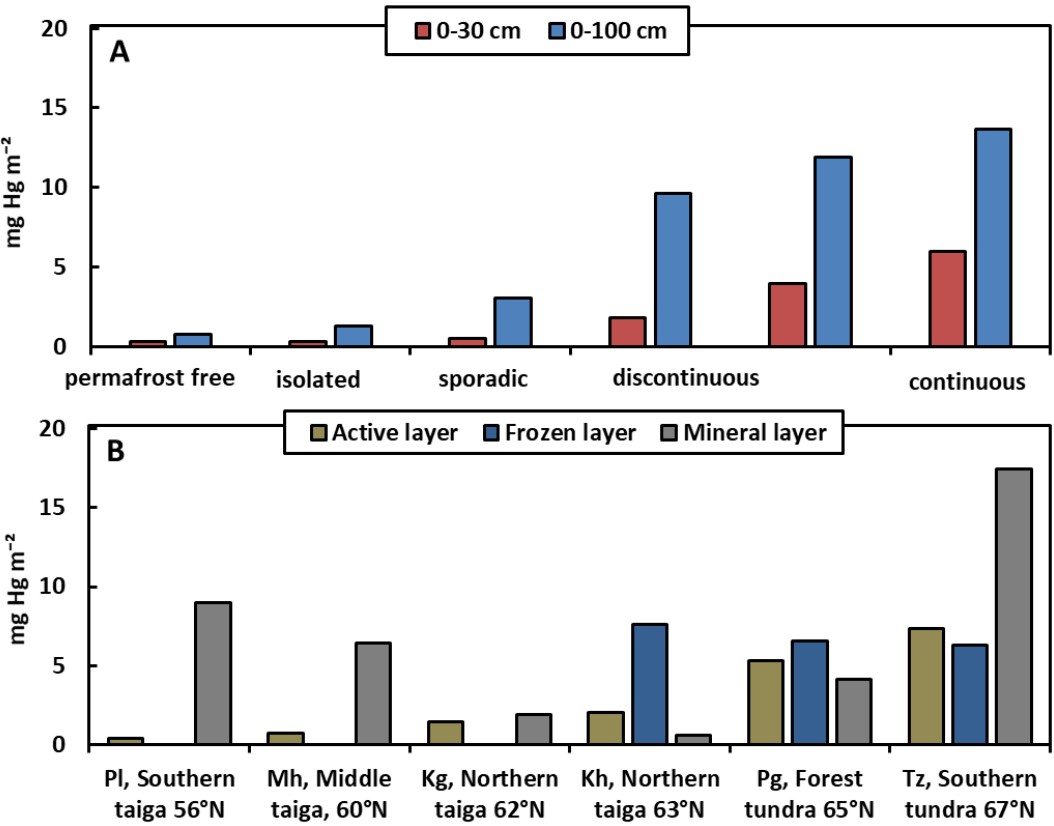

**Fig. 5.** Latitudinal variation in WSL soil Hg storage (mg Hg m$^{-2}$) in the 0-30 and 0-100 cm peat layer (**A**) and in the active, frozen and mineral layers (**B**). In the permafrost-free zone, the first 40 cm were used to calculate Hg storage in the active layer. The permafrost peat layer is fixed from the lower boundary of the active (unfrozen) layer down to 100 cm. Finally, for the mineral layer we considered only the first 10 cm below peat deposits across the full latitudinal gradient of the WSL peatland.






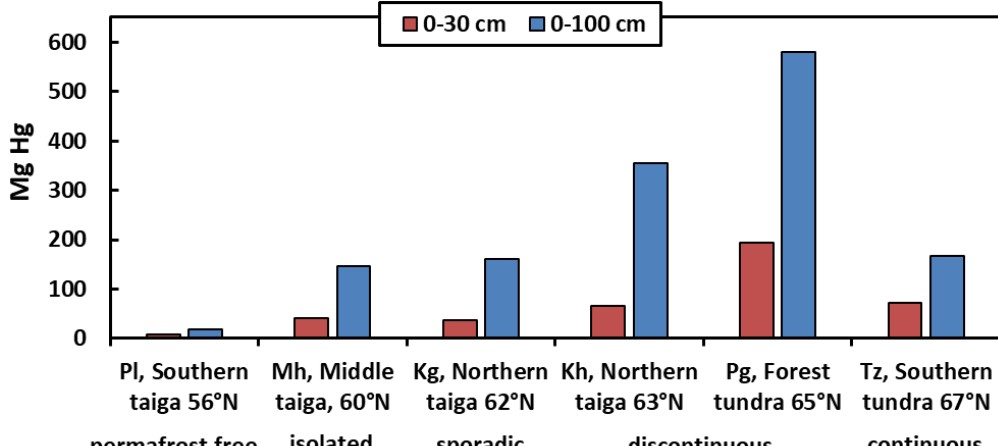




**Fig. 6.** Total, depth-integrated pools of Hg mass (Mg) in the upper 0-30 and 0-100 cm (red and
blue columns, respectively) of WSL frozen peatlands in each permafrost zone. The stocks are
calculated assuming the areal proportion of bogs from the landscape inventory across the WSL
(Sheng et al., 2004).



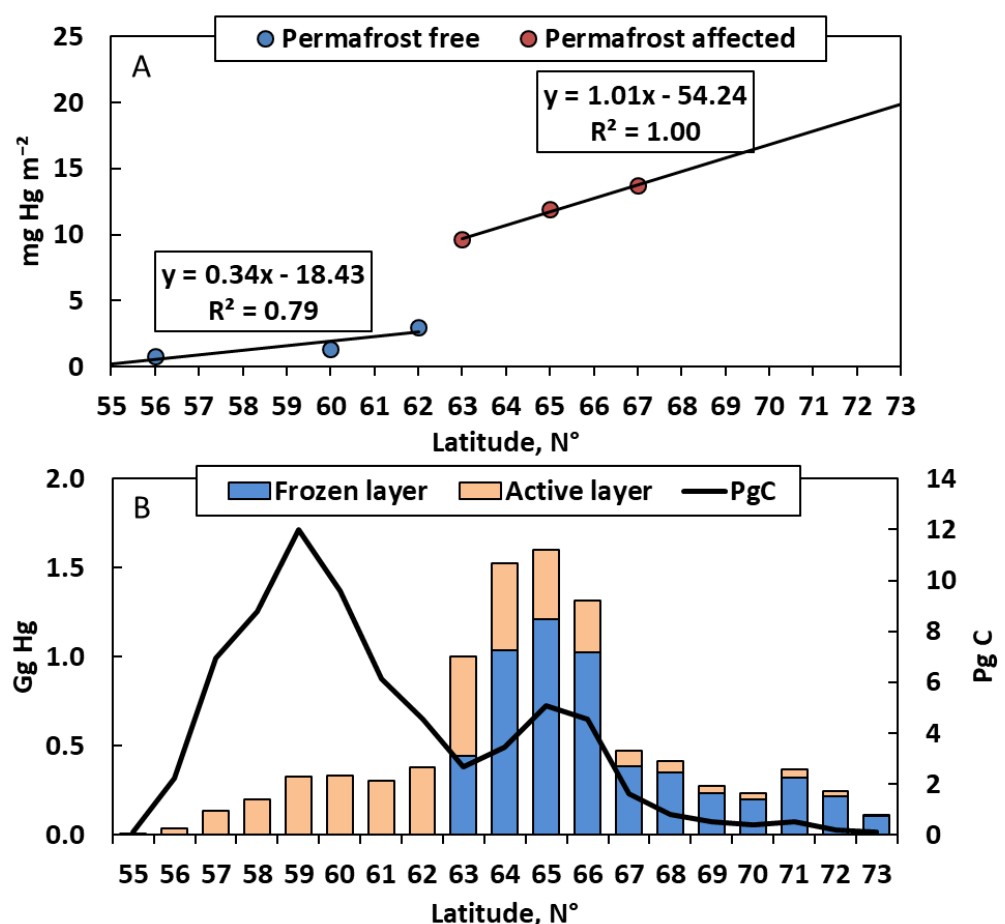




**Fig. 7**. Estimated Hg storage (mg Hg m$^{-2}$) in the 0-100 cm soil layer of the WSL (**A**) and
latitudinal distribution of the Hg pool (Gg) in active and frozen peat layers of the WSL (**B**).
Solid black line represents the WSL C pool (Pg) from Sheng et al. (2004).
















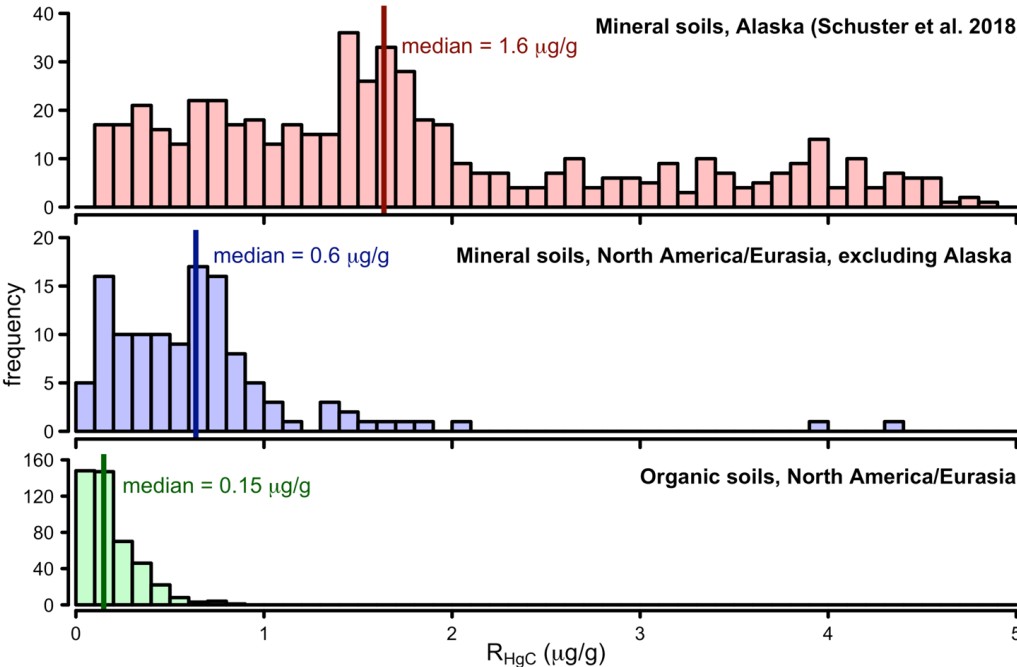




**Fig. 8**. Histogams of published and WSL $R_{HgC}$ data used for estimating the northern soil Hg pool: Alaskan mineral soils (top, Schuster et al., 2018), mineral soils from North America, excluding Alaska, and Eurasia (middle), and organic soils from North America and Eurasia, including Alaska (lower).










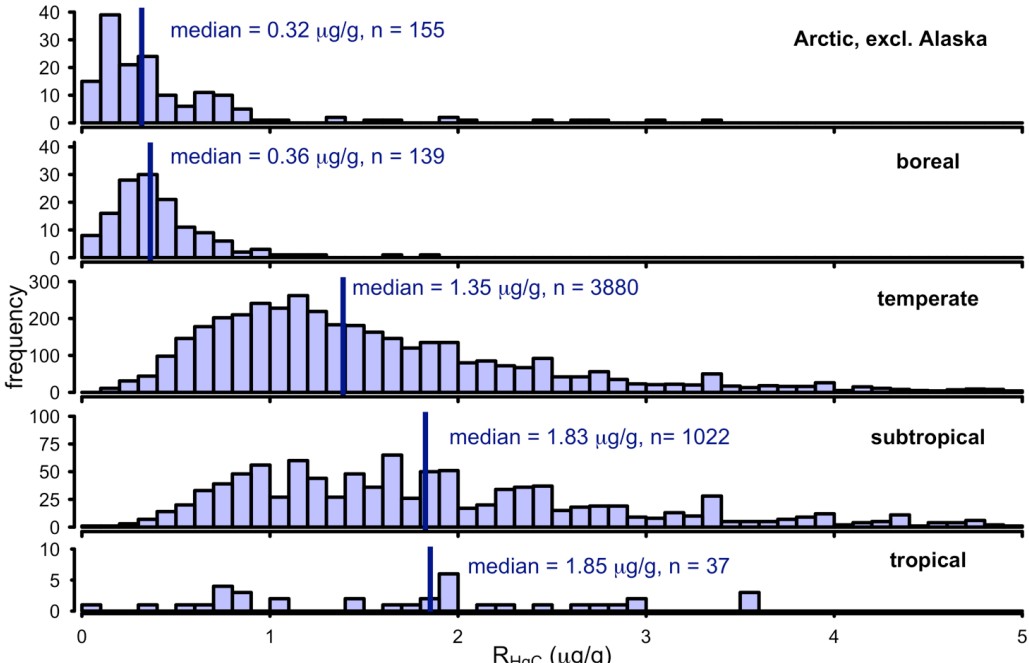

**Fig. 9.** Histograms and median $R_{HgC}$ for different global climate zones.