# Peer review of "A revised pan-Arctic permafrost soil Hg pool, based on western Siberia peat Hg and carbon observations"

_Biogeosciences, 2019_

## Referee Comment (RC1) · Anonymous Referee #1 · 4 Feb 2020

Lim and co-authors present a new set of peat observations from Siberia to improve estimates of mercury (Hg) storage in Arctic soils and permafrost. Their work fills important data gaps and will make a substantial contribution to the field. The manuscript is clear, well organized, and is supported by good figures. I recommend this paper for publication with minor revisions.

Specific comments:

Line 31: "Ocean" should be lower case.

Line 35: "arctic" should be capitalized.

[Figure]

Line 37: "Western" should be lower case.

Line 48: I'd replace "must be performed" with "are needed".

Line 78: Why is export most pronounced in the discontinuous permafrost zone?

Line 86: Spell out "North" in "N-America".

Line 116: You could omit "of the Supplementary Information". It's already implied by the "S" in Table S1.

Line 126: The abbreviation "MAAT" isn't helpful. Do you need it?

Line 374: "Ocean" should be lower case.

---

## Referee Comment (RC2) · Anonymous Referee #2 · 15 Mar 2020

Recent work by Schuster et al. (2018) and Olson et al. (2018) showed that arctic permafrost stores a significant amount of mercury (Hg), environmental toxicant harmful to human health and the environment. Climate change driven permafrost thaw will most likely lead to substantial Hg remobilization to the atmosphere and aquatic systems. In that context, a well constrained Hg budget in arctic permafrost is necessary. The two above-mentioned studies used Hg to carbon (Hg:C) ratios measured in Alaska, together with a northern soil C inventory, to estimate the amount of Hg stored in pan-Arctic northern soils. However, measurements of Hg:C ratios in Siberia are missing, hampering our ability to accurately estimate northern soil Hg pool. In this manuscript, Lim et al. report Hg and C concentrations, and Hg:C ratios, in six peat cores collected

in the Western Siberian Lowlands (WSL). Using these data, the authors revise the northern soil Hg pool to 557 Gg (0-300 cm), which is three times lower than the previous estimate of ∼1650 Gg by Schuster et al. (2018). Therefore, this manuscript will make an important contribution to the field after the authors address the following comments. Overall, I consider that the manuscript lacks precision in many aspects and the authors should clarify their Methods section.

1. Throughout the manuscript, the authors refer to northern soil Hg pools calculated by Schuster et al. (2018) et Olson et al. (2018) for the upper 1 m: 755 Gg and 184 Gg, respectively. Olson et al. (2018) actually showed that Arctic tundra soils store 184 Gg of Hg while boreal soils store additional 224 Gg. The authors therefore reported a pool of 408 Gg of Hg for northern tundra and boreal soils. Page 1068, Olson et al. say "Our combined estimate for Hg pools of 408 Gg for the top 100 cm of boreal and Arctic soils is about half of what Schuster et al. (2018) estimated was stored within upper soils". If the authors consider that 184 Gg is a better estimate and is a better comparison to the Schuster et al. study, please provide an explicit definition of "northern" soils to provide the readers an easier apple-to-apple comparison.

2. Throughout the manuscript, the authors suggest that according to Olson et al. (2018), the Hg:C ratio in Alaskan organic and mineral horizons ranges from 0.12 to 0.62 Gg/Pg. However, according to Table 1 in Olson et al. (2018), Hg:C ratios range from 0.27 Gg/Pg in organic soils to 0.62 Gg/Pg in mineral soils. Please edit the manuscript accordingly.

3. The authors extrapolate Eurasian soils Hg pool based on six peat cores collected in the WSL but do not discuss horizontal soil heterogeneity nor the need for additional samples in other parts of Siberia. I would appreciate a critical discussion on the soil sampling strategy used in this study. See Perkins et al. (2013) for tips. It is for instance usually recommended to implement a systematic sampling strategy or to combine replicate samples into a "composite sample".

4. According to section 2.2, C pools were multiplied with the respective Hg:C ratios for organic and mineral soils from north America (excluding Alaska) and Eurasia to estimate the northern soil Hg pool. I am not entirely sure what the authors mean by "excluding Alaska". Did they estimate the northern soil Hg pool by applying different Hg:C ratios for Alaska, or by simply assuming Alaska does not exist? Please clarify.

5. Page 14 and Figure 8, the authors suggest that "North American and Eurasian mineral soils Hg:C ratio was lower than Hg:C ratio reported for Alaska". Additionally, "the Hg:C ratio in organic soils was approximately 4 times lower than that in mineral soils of North America and Eurasia". I do not understand which dataset was used here. I would appreciate a table with the list of studies the authors are referring to. In lines 345-346 the authors mention "the literature data compilations of Olson et al. (2018) and Schuster et al. (2018)" but this is to my point of view not enough.

6. Same comment for the Hg:C ratios in various climate zones: which data were used? Again, I would really appreciate a table summarizing the literature used here. This entire section is too confusing as is.

7. The authors compare their 1084 Gg estimate of global Hg soil pool (0-30 cm) to the available literature. However, as mentioned by Outridge et al. (2018) (that should be cited here), most of these studies refer the amount of Hg in the actively recycling soil pool. For instance, the 950 Mg estimate by Outridge et al. (2018) refers to the top 10 cm. Similarly, Selin et al. (2008) refered to a layer ~ 15 cm deep.

Line-by-line comments:

Lines 38-39: "Hg concentrations increase from south to north in all soil horizons, reflecting enhanced net accumulation of atmospheric gaseous Hg by the vegetation Hg pump". As is, this sentence seems to suggest increasing vegetation uptake from south to north. However, as discussed in the manuscript, the Hg concentration increase is actually due to decreasing reemissions from south to north. Please edit this sentence accordingly (misleading as is).

Lines 70-71: see major comment #1.

Line 82: "strong year round net Hg(0) emission". Please clarify what you mean by "strong".

Line 91: "GIS" please define acronym.

Line 95: see major comment #2.

Line 126: please replace "atmospheric" by "ambient" and "increases" by "decreases".

Line 131: referring to the active layer as "unfrozen" soils is somewhat misleading since the active layer thaws during summer but freezes again in winter.

Lines 152-155: see major comment #3.

Lines 166-177: please define acronyms (BCR, MESS, NIST, SRM, ICP-MS).

Lines 187-189: unclear, see major comment #4.

Line 190: typo, "singe" should be "single".

Line 211 and throughout the manuscript: please use "PI" instead of the full name to make it easier to find the associated figure (same comment applies to all the sites).

Lines 218-224: how does this compare to other studies? Please strengthen the discussion.

Lines 225-229: how does this compare to other studies? Please strengthen the discussion.

Line 301: for consistency please use the same units throughout the manuscript (Gg/Pg).

Line 320: see major comment #2.

Lines 322-323: see major comment #1.

Lines 328-329: please add units for the medians.

Lines 318-352: I find this entire section confusing because I do not understand which data you are referring to. See major comment #5.

Lines 365-367: Please clarify which studies you are referring to. See major comment #6.

Lines 369-373: See major comment #7.

Figure 3: the caption should be self-explanatory. What do ALT, PF1 and PF2 mean?

References

Olson, C., M. Jiskra, H. Biester, J. Chow, and D. Obrist. 2018. "Mercury in Active-Layer Tundra Soils of Alaska: Concentrations, Pools, Origins, and Spatial Distribution." Global Biogeochemical Cycles 32 (7): 1058–73. https://doi.org/10.1029/2017GB005840.

Outridge, P. M., R. P. Mason, F. Wang, S. Guerrero, and L. E. Heimbürger-Boavida. 2018. "Updated Global and Oceanic Mercury Budgets for the United Nations Global Mercury Assessment 2018." Environmental Science & Technology 52 (20): 11466–77. https://doi.org/10.1021/acs.est.8b01246.

Perkins, Lora B., Robert R. Blank, Scot D. Ferguson, Dale W. Johnson, William C. Lindemann, and Ben M. Rau. 2013. "Quick Start Guide to Soil Methods for Ecologists." Perspectives in Plant Ecology, Evolution and Systematics 15 (4): 237–44. https://doi.org/10.1016/j.ppees.2013.05.004.

Schuster, Paul F., Kevin M. Schaefer, George R. Aiken, Ronald C. Antweiler, John F. Dewild, Joshua D. Gryziec, Alessio Gusmeroli, et al. 2018. "Permafrost Stores a Globally Significant Amount of Mercury." Geophysical Research Letters 45 (3): 2017GL075571. https://doi.org/10.1002/2017GL075571.

Selin, Noelle E., Daniel J. Jacob, Robert M. Yantosca, Sarah Strode, Lyatt Jaeglé,

and Elsie M. Sunderland. 2008. "Global 3-D Land-Ocean-Atmosphere Model for Mercury: Present-Day versus Preindustrial Cycles and Anthropogenic Enrichment Factors for Deposition." Global Biogeochemical Cycles 22 (2): GB2011. https://doi.org/10.1029/2007GB003040.

---

## Author Comment (AC2) · 9 Apr 2020

Anonymous Referee #2 Overall assessment. Recent work by Schuster et al. (2018) and Olson et al. (2018) showed that arctic permafrost stores a significant amount of mercury (Hg), environmental toxicant harmful to human health and the environment. Climate change driven permafrost thaw will most likely lead to substantial Hg remobilization to the atmosphere and aquatic systems. In that context, a well constrained Hg budget in arctic permafrost is necessary. The two above-mentioned studies used Hg to carbon (Hg:C) ratios measured in Alaska, together with a northern soil C inventory, to estimate the amount of Hg stored in pan-Arctic northern soils. However,

measurements of Hg:C ratios in Siberia are missing, hampering our ability to accurately estimate northern soil Hg pool. In this manuscript, Lim et al. report Hg and C concentrations, and Hg:C ratios, in six peat cores collected in the Western Siberian Lowlands (WSL). Using these data, the authors revise the northern soil Hg pool to 557 Gg (0-300 cm), which is three times lower than the previous estimate of _1650 Gg by Schuster et al. (2018). Therefore, this manuscript will make an important contribution to the field after the authors address the following comments. Overall, I consider that the manuscript lacks precision in many aspects and the authors should clarify their Methods section. - Response: we clarified the definitions of terms and methods and explained the sampling strategy.

Comment 1. Throughout the manuscript, the authors refer to northern soil Hg pools calculated by Schuster et al. (2018) et Olson et al. (2018) for the upper 1 m: 755 Gg and 184 Gg, respectively. Olson et al. (2018) actually showed that Arctic tundra soils store 184 Gg of Hg while boreal soils store additional 224 Gg. The authors therefore reported a pool of 408 Gg of Hg for northern tundra and boreal soils. Page 1068, Olson et al. say "Our combined estimate for Hg pools of 408 Gg for the top 100 cm of boreal and Arctic soils is about half of what Schuster et al. (2018) estimated was stored within upper soils". If the authors consider that 184 Gg is a better estimate and is a better comparison to the Schuster et al. study, please provide an explicit definition of "northern" soils to provide the readers an easier apple-to-apple comparison.

Reply: We agree with the reviewer that there is some confusion on the extent of the area for which the pool estimates apply. We change the nomenclature from "northern" soils to "northern circumpolar permafrost region". We thereby rely on the definition by Hugelius et al. 2014 (Biogeosciences) Please note that this definition does not include boreal soils, for which we calculate a Hg inventory separately, e.g. see Table 4 and Figure 9. Olson et al. also calculated Hg soil pools separately for the northern circumpolar permafrost region and for the boreal region and the 408 Gg reported by the reviewer above sum up the two regions. In our comparison of the three studies

(Schuster et al. 2018, Olson et al. 2018 and this contribution) we deal with northern circumpolar permafrost region only, which is based on the definition of Hugelius et al. 2014 in all three studies. Differences in Hg pool estimates between the three different studies originate from different RHgC ratios applied to the same carbon pool estimate rather than from the area.

Comment 2. Throughout the manuscript, the authors suggest that according to Olson et al. (2018), the Hg:C ratio in Alaskan organic and mineral horizons ranges from 0.12 to 0.62 Gg/Pg. However, according to Table 1 in Olson et al. (2018), Hg:C ratios range from 0.27 Gg/Pg in organic soils to 0.62 Gg/Pg in mineral soils. Please edit the manuscript accordingly.

Reply: Please note that we discovered an error in Olson et al.: in Olsen'18 Table 1, and main text the median Hg:C ratio for organic soils is indicated to be 0.274 Pg/Gg; yet the IQR is 95-193 Gg/Pg. Also, multiplying the 274 number by the carbon pool (217 Pg) does not giveyield the 26 Gg Hg pool. The correct median Hg:C ratio for organic soils should be 0.119 Gg/Pg. This error did not affect the final pool size calculation in Olsen et al.; it is just an error in text and table. We added a line to our MS to indicate this error (L334, and 454):

"Note that Olson et al. Table 1 has an incorrect organic soil RHgC of 0.274 Gg Pg-1, which should be 0.119 Gg Pg-1; the typo did not affect their soil Hg budgets."

Comment 3. The authors extrapolate Eurasian soils Hg pool based on six peat cores collected in the WSL but do not discuss horizontal soil heterogeneity nor the need for additional samples in other parts of Siberia. I would appreciate a critical discussion on the soil sampling strategy used in this study. See Perkins et al. (2013) for tips. It is for instance usually recommended to implement a systematic sampling strategy or to combine replicate samples into a "composite sample".

Response: We understand and share the reviewer's concern. There are multiple reasons for rather limited sampling volume in our work: 1. Sampling strategy aimed to

retrieve intact cores, so that 14C dating and C/Hg stable isotope analysis (ongoing) will help assess C remineralization rates, and Hg deposition/re-emission. Composite sampling from multiple cores, at the same depths perturbs these objectives. Ideally we would take 5 peat cores per permafrost region, in order to understand intra-site, local variability in all signals. With the current funding rates in France, Europe, and Russia (<10%) this is vary challenging, in particular because field logistics in Russia and sample transport demand substantial financial resources; give us a bag of money and we will get those 25 cores :). We added the following text on sampling strategy to the methods section: "Field logistics and financial support did not make it possible to study multiple cores from each climate zone." In response to this comment, we will emphasize the need for additional work in Eastern Siberia in the abstract and discussion.

Overall, we fully acknowledge the limitations of the RHgC upscaling approach and we understand a need to move towards a spatially resolved Hg pool estimate to incorporate heterogeneity in geogenic Hg and soil formation and to achieve a more detailed analysis of risk areas with respect to Hg transfer to aquatic ecosystems and evasion to the atmosphere Furthermore, we estimated the lateral variability in trace (toxic) metal concentration in peat cores from various micro-landscapes (mound, depression) in the same permafrost zone (latitude) based on our former work of elementary composition of peat across the WSL (Stepanova et al., 2015, Appl. Geochemistry, 53, 53–70, doi:10.1016/j.apgeochem.2014.12.004, Fig. 5, and associated Supplementary Information). These variations for Cd and Pb concentrations range from 25 to 50%. Similar range is exhibited by Fe and P. Although these elements cannot serve as straightforward analogues to highly labile Hg, we believe that the lateral variations in Hg concentration should be within the IQ range of Hg:C ratio as depicted in Fig. 4 of our manuscript, and as such, these variations do not sizably affect the overall estimation of Hg pools in Eurasian peat soils.

Comment 4. According to section 2.2, C pools were multiplied with the respective Hg:C ratios for organic and mineral soils from north America (excluding Alaska) and Eurasia

to estimate the northern soil Hg pool. I am not entirely sure what the authors mean by "excluding Alaska". Did they estimate the northern soil Hg pool by applying different Hg:C ratios for Alaska, or by simply assuming Alaska does not exist? Please clarify.

Response: We acknowledge that Alaska still exists! But we did not include Alaskan mineral soil Hg:C ratios in our estimate of the mineral soil Hg:C ratio representative for the entire northern circumpolar permafrost region. The reason is that the elevated Alaskan mineral soil Hg:C ratio is biased high and not representative of the large Siberian mineral soils. On Lines 360-362 of the discussion paper we quantify the systematic error made by "excluding Alaska": "The error made by neglecting high RHgC in Alaskan mineral soils is small, on the order of 2.5 Gg Hg, as estimated from the relatively small Alaskan C pool of 2.6 Pg C (Tarnocai et al., 2009)." The 0-3 m Hg pool in the northern circumpolar permafrost region is 557 Gg with an interquartile range between 371 and 699 Gg. We therefore argue that a systematic underestimation in the order of 2.5 Gg (approx. 0.5% of the total pool) is negligible given the large uncertainties associated with the estimate. The carbon pool estimates for different soil types from Hugelius et al (2014) are for the entire northern circumpolar permafrost region and no such data are available for Alaska on a soil type level. Therefore we are not able to provide a more accurate estimate at this stage.

Comment 5. Page 14 and Figure 8, the authors suggest that "North American and Eurasian mineral soils Hg:C ratio was lower than Hg:C ratio reported for Alaska". Additionally, "the Hg:C ratio in organic soils was approximately 4 times lower than that in mineral soils of North America and Eurasia". I do not understand which dataset was used here. I would appreciate a table with the list of studies the authors are referring to. In lines 345-346 the authors mention "the literature data compilations of Olson et al. (2018) and Schuster et al. (2018)" but this is to my point of view not enough.

Response: For our analysis we combine four datasets: the original permafrost Hg data from Schuster et al. 2018 (ca. 590 datapoints), a global compilation of Hg soil data by Schuster et al (ca. 11000 datapoints) a dataset of permafrost-affected Arctic and

Boreal Hg soil data used by Olson et al. 2018 (958 datapoints) and the original soil data from the western Siberian lowlands (223 datapoints). We refer to the data availability statement, where we provide a link to the data sources used in this analysis. The dataset original in this study is provided in the supporting information of this study. The dataset used in Olson et al 2018 is currently not publically available and has to be acquired by contacting the corresponding author.

Comment 6. Same comment for the Hg:C ratios in various climate zones: which data were used? Again, I would really appreciate a table summarizing the literature used here. This entire section is too confusing as is.

Response: We refer to the data availability statement, where we provide a link to the data sources used in this analysis. Data availability. Hg and C concentration data of the WSL soil samples are available in the supplement. The permafrost data from Schuster et al. 2018 and a global compilation of RHgC data is available as supplementary information (https://agupubs.onlinelibrary.wiley.com/doi/full/10.1002/2017GL075571, last access: 6 December 2019). The Arctic and boreal soil data from the Olson et al. 2018 study is available from the corresponding author upon request. Note that Olson et al. Table 1 has an incorrect organic soil RHgC of 0.274 Gg Pg-1, which should be 0.119 Gg Pg-1. The data from the tropical climate zone was compiled from original publications of Almeida (2005); Almeida et al. (2005); Campbell et al. (2003); Melendez-Perez et al. (2014). We now added these reference to the main text as well.

Comment 7. The authors compare their 1084 Gg estimate of global Hg soil pool (0-30 cm) to the available literature. However, as mentioned by Outridge et al. (2018) (that should be cited here), most of these studies refer the amount of Hg in the actively recycling soil pool. For instance, the 950 Mg estimate by Outridge et al. (2018) refers to the top 10 cm. Similarly, Selin et al. (2008) referred to a layer _ 15 cm deep.

Response: We agree that the soil depth intervals to estimate the Hg pools in soils varies between different studies and thereby contributes to the large range in pool

estimates. In the revised MS we will report the soil depth intervals of individual studies. The depth of 0 to 30 cm has been used in our study because this interval is established in the carbon community and soil carbon inventories exist for this depth range. We do not interpret this depth interval as the soil Hg pool that is actively recycled. Such a simplification would not take into account the heterogeneity between different soil types and the complexity of Hg cycling in soils. We will add the Outridge et al. 2018 reference, but we could not find where the 950 Gg soil pool estimate for the 0-10cm is referred to. In Table 1 of the Outridge et al. (2018) paper, the soil Hg pool (described as organic layers) is estimated to be 150 Gg, but no depth interval is given in their work.

Line-by-line comments:

Lines 38-39: "Hg concentrations increase from south to north in all soil horizons, reflecting enhanced net accumulation of atmospheric gaseous Hg by the vegetation Hg pump". As is, this sentence seems to suggest increasing vegetation uptake from south to north. However, as discussed in the manuscript, the Hg concentration increase is actually due to decreasing reemissions from south to north. Please edit this sentence accordingly (misleading as is).

Thanks for pointing out this inconsistency. We changed the phrase as follows: "Hg concentrations increase from south to north in all soil horizons, reflecting a higher stability of sequestered Hg with respect to re-emission."

Lines 70-71: see major comment #1. - We agree and changed the nomenclature accordingly.

Line 82: "strong year round net Hg(0) emission". Please clarify what you mean by "strong". - Deleted the word 'strong'

Line 91: "GIS" please define acronym. - Deleted the term 'GIS'; it is not critical; expanding would lengthen the phrase unnecessarily.

Line 95: see major comment #2. -Necessary explanation is added to the manuscript.

Line 126: please replace "atmospheric" by "ambient" and "increases" by "decreases". -Changed as suggested

Line 131: referring to the active layer as "unfrozen" soils is somewhat misleading since the active layer thaws during summer but freezes again in winter. - We deleted the word "unfrozen"

Lines 152-155: see major comment #3. We agree and emphasized the need for additional work in Eastern Siberia. We also note that the lateral variations in trace metal concentrations in peatbogs of WSL (mound vs depression) are within 30-50% (based on our previous work on elementary composition of peat profiles). These variations are within the IQR of recommended values and thus should not affect the overall assessment of Hg pools.

Lines 166-177: please define acronyms (BCR, MESS, NIST, SRM, ICP-MS). - It is fairly uncommon to fully write out acronyms of reference materials BCR, MESS, NIST SRM, i.e. doing a google on the acronyms (with reference number) will lead to the right information; doing a google on full terms will not. We expanded the term ICP-MS. SRM was deleted.

Lines 187-189: unclear, see major comment #4. -The carbon pool estimates for different soil types from Hugelius et al (2014) are for the entire northern circumpolar permafrost region and no such data are available for Alaska on a soil type level.

Line 190: typo, "singe" should be "single". - Typo corrected

Line 211 and throughout the manuscript: please use "Pl" instead of the full name to make it easier to find the associated figure (same comment applies to all the sites). - Well, that would add another 6 acronyms to the MS; we prefer to discuss sites by naming them fully; Note that the Figure captions include both full names and abbreviation, such as Plotnikovo (Pl).

Lines 218-224: how does this compare to other studies? Please strengthen the discussion. - Please note that we separated the results and discussion part in this manuscript. In section 3.1. we present the results from the peat cores sampled along the western Siberian lowlands. A comparison with other studies is given in discussion section 4.2.

Lines 225-229: how does this compare to other studies? Please strengthen the discussion. - Please note that we separated the results and discussion part in this manuscript. In section 3.1. we present the results from the peat cores sampled along the western Siberian lowlands. A comparison with other studies is given in section 4.2.

Line 301: for consistency please use the same units throughout the manuscript (Gg/Pg). - Corrected throughout the MS

Line 320: see major comment #2. - Necessary edits and corrections were applied.

Lines 322-323: see major comment #1. - We agree and changed the nomenclature accordingly.

Lines 328-329: please add units for the medians. - The units were added

Lines 318-352: I find this entire section confusing because I do not understand which data you are referring to. See major comment #5. - We clarified as much as possible, adding "Dalton Highway, Noatak Natinoal Preserve, 8 Mile Lake Observatory" localities to the Olson et al., site description.

Lines 365-367: Please clarify which studies you are referring to. See major comment #6. - We revised the data availability statement and added relevant references. We added references from the Data availability Statement to the main text now :" (Campbell et al., 2003; Almeida, 2005; Almeida et al. 2005; Melendez-Perez et al., 2014; Olson et al., 2018)"

Lines 369-373: See major comment #7. - We explained specific soil depths with relevant references.

Figure 3: the caption should be self-explanatory. What do ALT, PF1 and PF2 mean?

The ALT stands for Active Layer Thickness and PF1 and PF2 designate surface and deep permafrost layers. See Table 1 for exact abbreviations of ALT, PF1 and PF2. Specifically, PF1 is frozen peat, (ALT-100 ÑĄÐij); PF2 is frozen peat (ALT to mineral layer). We added a pertinent reference to Table 1, where these abbreviations are presented.

References

Olson, C., M. Jiskra, H. Biester, J. Chow, and D. Obrist. 2018. "Mercury in Active-Layer Tundra Soils of Alaska: Concentrations, Pools, Origins, and Spatial Distribution." Global Biogeochemical Cycles 32 (7): 1058–73. https://doi.org/10.1029/2017GB005840. Outridge, P. M., R. P. Mason, F. Wang, S. Guerrero, and L. E. HeimbuÌLrger-Boavida. 2018. "Updated Global and Oceanic Mercury Budgets for the United Nations Global Mercury Assessment 2018." Environmental Science & Technology 52 (20): 11466–77. https://doi.org/10.1021/acs.est.8b01246. Perkins, Lora B., Robert R. Blank, Scot D. Ferguson, Dale W. Johnson, William C. Lindemann, and Ben M. Rau. 2013. "Quick Start Guide to Soil Methods for Ecologists." Perspectives in Plant Ecology, Evolution and Systematics 15 (4): 237–44. https://doi.org/10.1016/j.ppees.2013.05.004. Schuster, Paul F., Kevin M. Schaefer, George R. Aiken, Ronald C. Antweiler, John F. Dewild, Joshua D. Gryziec, Alessio Gusmeroli, et al. 2018. "Permafrost Stores a Globally Significant Amount of Mercury." Geophysical Research Letters 45 (3): 2017GL075571. https://doi.org/10.1002/2017GL075571. Selin, Noelle E., Daniel J. Jacob, Robert M. Yantosca, Sarah Strode, Lyatt Jaeglé, and Elsie M. Sunderland. 2008. "Global 3-D Land-Ocean-Atmosphere Model for Mercury: Present-Day versus Preindustrial Cycles and Anthropogenic Enrichment Factors for Deposition." Global Biogeochemical Cycles 22 (2): GB2011. https://doi.org/10.1029/2007GB003040.

We thank the reviewer # 2 for very insightful suggestions.

---

## Author Comment (AC3) · 9 Apr 2020

In order to further address the comment No 3 of Reviewer 2, we compared our unpublished data on elementary composition of peat with those of Raudina et al. (2019) and Stepanova et al. (2015) for the same key areas. We used a non-parametric Mann-Whitney U test for paired data at a significance level of 0.05 to assess the difference between sites (micro-landscapes) for each key area. The overwhelming majority of elements do not exhibit statistically significant differences between different peat cores. Thus, in the middle taiga region (Mukhrino), only Gd and Tb were sizably different. In Khanymey, only Mg showed statistically significant difference between different peat

cores. In the forest tundra of Pangody, only Na, Ti, As, Cd, Tl, Pb exhibited sizable differences. Finally, various peat cores from southern tundra (Tazovsky) differed only in the concentration of Ca, Ni, Cu, Mo and Hf.

Taken together, we believe that one single core is sufficiently representative for the purpose of assessment of both elementary composition and overall stock of elements (including Hg).

Table R1. Comparison of concentrations of major and trace elements in peat cores from different microlandscapes (mound, depression) of WSL peatbogs studied in this work (and our unpublished data) with results of Raudina et al. (2019); Stepanova et al. (2015). Only the elements exhibiting statistically significant differences are presented. See attachement.

References:

Raudina, T. V. and Loiko, S. V.: Properties and major element concentrations in peat profiles of the polygonal frozen bog in Western Siberia, in: IOP Conference Series: Earth and Environmental Science (Vol. 400, No. 1, p. 012009). IOP Publishing. (2019, November).

Stepanova, V. A., Pokrovsky, O. S., Viers, J., Mironycheva-Tokareva, N. P., Kosykh, N. P., and Vishnyakova, E. K.: Elemental composition of peat profiles in western Siberia: Effect of the micro-landscape, latitude position and permafrost coverage, Appl. Geochemistry, 53, 53–70, doi:10.1016/j.apgeochem.2014.12.004, 2015.

**Table R1.** Comparison of concentrations of major and trace elements in peat cores from different microlandscapes (mound, depression) of WSL peatbogs studied in this work (and our unpublished data) with results of Raudina et al. (2019); Stepanova et al. (2015). Only the elements exhibiting statistically significant differences are presented.

| Elements | U | Z | p-value |
|---|---|---|---|
| Stepanova et al., 2015 Mukhrino, Middle taiga | | | |
| Gd | 6 | -3.3 | 0.001 |
| Tb | 0 | 3.5 | 0.000 |
| Our unpublished data Khanymey, Northern taiga | | | |
| Mg | 19 | 2.6 | 0.009 |
| Stepanova et al., 2015 Pangody, Forest tundra | | | |
| Na | 0 | 2.4 | 0.016 |
| Ti | 2 | -2.0 | 0.042 |
| As | 1 | -2.2 | 0.027 |
| Cd | 0 | -2.4 | 0.016 |
| Tl | 1 | -2.2 | 0.027 |
| Pb | 2 | -2.0 | 0.042 |
| Raudina et al., 2019 | | | |
| Ca | 20 | 2.4 | 0.017 |
| Ni | 2 | 3.7 | 0.000 |
| Cu | 22 | 2.2 | 0.025 |
| Mo | 17 | 2.6 | 0.009 |
| Hf | 16 | 2.7 | 0.008 |

**Fig. 1.** Table R1. Comparison of concentrations of major and trace elements in peat cores from different microlandscapes (mound, depression) of WSL peatbogs

---

## Author Response (AR1)

Dear Dr Andreas Richter

Our manuscript received two very constructive comments and we carefully revised the manuscript following their remarks and suggestions. We thank you for giving us an opportunity to revise this paper.

Answer to Associate Editor:

**You correctly noted that we have not provided a clear and adequate response to comment 3 of reviewer 2.** In fact we figured out that only part of his/her comment was addressed and we present below our detailed response to this comment.

**You also underlined that "..the reviewer made a very important point here in that conclusions on the Eurasian permafrost Hg pool cannot be drawn from 6 peat cores alone" and requested that we discuss the limitations of our approach clearly in the revised manuscript.**
Here, we would like to highlight a misunderstanding by the reviewer concerning the suggested extrapolation to Eurasian soils, for which we do not provide an estimate. We estimate the WSL Hg pool and the pan-arctic Hg pool (but not the Eurasian pool). The 223 data points from the six cores along the WSL transect sampled and reported the first time in this study were the sole basis for the estimate of the WSL Hg pool (Section 3.2). For the estimation of the Panarctic permafrost soil pool we included a broader dataset of 131 mineral soil samples and 449 organic soil samples. This dataset covers Europe and North America. The WSL soil samples therefore accounted only for approximately 30% of the entire dataset used to assess the panarctic permafrost soil pool. We added a table in the supporting information (Table S4) providing an overview of the data used for the pan-arctic assessment of RHG. We also slightly ($< 5\%$) revised our numbers as we found a new reference on Hg in peat cores from Northeastern European Russia.

Finally, we would like to sincerely apologize for our rather cynical response ("Give us a bag of money and we will get those 25 cores :)"). Correct answer is that field logistics and financial support did not make it possible to study multiple cores from each climate zone of the WSL and each region of Siberia. We deleted the sentences concerning the funding situation in the final response.

**Answers to Reviewers**

**Referee #1**

Lim and co-authors present a new set of peat observations from Siberia to improve estimates of mercury (Hg) storage in Arctic soils and permafrost. Their work fills important data gaps and will make a substantial contribution to the field. The manuscript is clear, well organized, and is supported by good figures. I recommend this paper for publication with minor revisions.

Specific comments:

**Line 31: "Ocean" should be lower case.**
Changed as suggested

**Line 35: "arctic" should be capitalized.**
Changed as suggested

**Line 37: "Western" should be lower case.**
Changed as suggested

**Line 48: I'd replace "must be performed" with "are needed".**
Changed as suggested

**Line 78: Why is export most pronounced in the discontinuous permafrost zone?**
This is where enhanced thawing exposes fresh soil organic matter (OM) to relatively important summer temperature increases, and naturally important run-off. Furthermore, the active layer depth is high in this region and this helps to excavates large amount of OM from deeper soil horizons. We added a comment to the phrase, clarifying this: "…due to thawing of fresh soil organic matter and maximal active layer depth…" (**L 80-81**)

**Line 86: Spell out "North" in "N-America".**
Changed as suggested

**Line 116: You could omit "of the Supplementary Information". It's already implied by the "S" in Table S1.**
Changed as suggested

**Line 126: The abbreviation "MAAT" isn't helpful. Do you need it?**
We use the abbreviation MAAT in Lines 400 and 401, and therefore prefer to keep it

**Line 374: "Ocean" should be lower case.**
Changed as suggested

**Referee #2**

**Overall assessment.**

**Recent work by Schuster et al. (2018) and Olson et al. (2018) showed that arctic permafrost stores a significant amount of mercury (Hg), environmental toxicant harmful to human health and the environment. Climate change driven permafrost thaw will most likely lead to substantial Hg remobilization to the atmosphere and aquatic systems. In that context, a well constrained Hg budget in arctic permafrost is necessary. The two above-mentioned studies used Hg to carbon (Hg:C) ratios measured in Alaska, together with a northern soil C inventory, to estimate the amount of Hg stored in pan-Arctic northern soils. However, measurements of Hg:C ratios in Siberia are missing, hampering our ability to accurately estimate northern soil Hg pool. In this manuscript, Lim et al. report Hg and C concentrations, and Hg:C ratios, in six peat cores collected in the Western Siberian Lowlands (WSL). Using these data, the authors revise the northern soil Hg pool to 557 Gg (0-300 cm), which is three times lower than the previous estimate of _1650 Gg by Schuster et al. (2018). Therefore, this manuscript will make an important contribution to the field after the authors address the following comments. Overall, I consider that the manuscript lacks precision in many aspects and the authors should clarify their Methods section.**

We clarified the definitions of terms and methods and explained the sampling strategy. We would like to underline that our revised northern soil Hg pool estimation originate from both new data on the WSL territory acquired in this study (approx.. 30%) and available literature information (remaining 70%).

**Comment 1. Throughout the manuscript, the authors refer to northern soil Hg pools calculated by Schuster et al. (2018) et Olson et al. (2018) for the upper 1 m: 755 Gg and 184 Gg, respectively. Olson et al. (2018) actually showed that Arctic tundra soils store 184 Gg of Hg while boreal soils store additional 224 Gg. The authors therefore reported a pool of 408 Gg of Hg for northern tundra and boreal soils. Page 1068, Olson et al. say "Our combined estimate for Hg pools of 408 Gg for the top 100 cm of boreal and Arctic soils is about half of what Schuster et al. (2018) estimated was stored within upper soils". If the authors consider that 184 Gg is a better estimate and is a better comparison to the Schuster et al. study, please provide an explicit definition of "northern" soils to provide the readers an easier apple-to-apple comparison.**

We agree with the reviewer that there is some confusion on the extent of the area for which the pool estimates apply. We change the nomenclature from "northern" soils to "pan-Arctic permafrost soils". We thereby rely on the definition for the northern circumpolar permafrost region by Hugelius et al. 2014 (Biogeosciences).

Please note that this definition does not include boreal soils, for which we calculate a Hg inventory separately, e.g. see Table 4 and Figure 9. Olson et al. also calculated Hg soil pools separately for the northern circumpolar permafrost region and for the boreal region and the 408 Gg reported by the reviewer above sum up the two regions. In our comparison of the three studies (Schuster et al. 2018, Olson et al. 2018 and this contribution) we deal with northern circumpolar permafrost region only, which is based on the definition of Hugelius et al. 2014 in all three studies. Differences in Hg pool estimates between the three different studies originate from different $R_{HgC}$ ratios applied to the same carbon pool estimate rather than from the area.

**Comment 2.**

**Throughout the manuscript, the authors suggest that according to Olson et al. (2018), the Hg:C ratio in Alaskan organic and mineral horizons ranges from 0.12 to 0.62 Gg/Pg. However, according to Table 1 in Olson et al. (2018), Hg:C ratios range from 0.27 Gg/Pg in organic soils to 0.62 Gg/Pg in mineral soils. Please edit the manuscript accordingly.**

Please note that we discovered an error in Olson et al.: in Olsen'18 Table 1, and main text the median Hg:C ratio for organic soils is indicated to be 0.274 Gg/Pg; yet the IQR is 95-193 Gg/Pg. Also, multiplying the 274 number by the carbon pool (217 Pg) does not yield the 26 Gg Hg pool. The correct median Hg:C ratio for organic soils should be 0.119 Gg/Pg. This error did not affect the final pool size calculation in Olsen et al.; it is just an error in text and table. We added a line to our MS to indicate this error (L322 and 439): "Note that Olson et al. Table 1 has an incorrect organic soil $R_{HgC}$ of 0.274 Gg Pg$^{-1}$, which should be 0.119 Gg Pg$^{-1}$; the typo did not affect their soil Hg budgets."

**Comment 3.**

**The authors extrapolate Eurasian soils Hg pool based on six peat cores collected in the WSL but do not discuss horizontal soil heterogeneity nor the need for additional samples in other parts of Siberia. I would appreciate a critical discussion on the soil sampling strategy used in this study. See Perkins et al. (2013) for tips. It is for instance usually recommended to implement a systematic sampling strategy or to combine replicate samples into a "composite sample".**

We understand and share the reviewer's concern. There are multiple reasons for rather limited sampling volume of the WSL data. First, our sampling strategy aimed to retrieve intact cores, so that $^{14}$C dating and C/Hg stable isotope analysis (ongoing) will help assess C remineralization rates, and Hg deposition/re-emission. Composite sampling from multiple cores at the same depths perturbs these objectives. Ideally we would take 5 peat cores per permafrost region, in order to understand intra-site, local variability in all signals. We added the following text on sampling strategy to the methods section: "Field logistics and financial support did not make it possible to study multiple cores from each climate zone."
In response to this comment, we emphasized the need for additional work in Eastern Siberia in the abstract (L 48-49), discussion (L 359-361), and conclusions (L 431-432).

Second, we would like to highlight a misunderstanding concerning the extrapolation to Eurasian soils, for which we do not provide an estimate. We estimate the WSL Hg pool and the pan-arctic Hg pool (but not the Eurasian pool). The 223 datapoints from the six cores along the WSL transect sampled and reported the first time in this study were the sole basis for the estimate of the WSL Hg pool (Section 3.2). For the estimation of the Panarctic permafrost soil pool we included a broader dataset of 131 mineral soil samples and 449 organic soil samples. These dataset cover Europe and North America (excluding samples from Alaska). Moreover, to generate Fig. 9 and Table 4, we combined four datasets: the original permafrost Hg data from Schuster et al. 2018 (ca. 590 data points), a global compilation of Hg soil data by Schuster et al (ca. 11000 data points), a dataset of permafrost-affected Arctic and Boreal Hg soil data used by Olson et al. 2018 (958 data points) and the original soil data from the western Siberian lowlands (223 data points). The new WSL soil samples therefore accounted only for 2% of the entire database used to assess the Hg soil pool. We added a new table in the supporting information (Table S4) providing an overview of the data used for the pan-arctic assessment of $R_{HgC}$.

We also found an additional reference from Northeastern European Russia reporting Hg and C data for peatlands which corroborate our results from the WSL (Vasilevich et al. 2018). We added the data from (Vasilevich et al. 2018) to our database and also moved the southernmost site of the WSL (Plotnikovo (Pl)) which is located in the absent permafrost region

to the boreal biome. As a result the calculated RHgC and pool sizes of the Panarctic Permafrost Pool and the boral Biome changes slightly (<5%) in the revised version of the manuscript. The changes have no implications to the overall interpretation and conclusions of the manuscript.

Overall, we fully acknowledge the limitations of the RHgC upscaling approach and we understand a need to move towards a spatially resolved Hg pool estimate to incorporate horizontal soil heterogeneity in geogenic Hg and soil formation. In the discussion version of the paper we have emphasized the need for more studies in the Abstract as follows: "Additional soil and river studies must be performed in Eastern and Northern Siberia to lower the uncertainty on these estimates, and assess the timing of Hg release to atmosphere and rivers." In section 4.2, we have discussed the variability in soils with respect to the applied RHgC approach as follows: "In particular, turbel and orthel mineral soils, which are estimated to contain 49 to 62% of total arctic C (Hugelius et al., 2014) and 36 to 85% of Hg at the various depth intervals need to be further investigated."

In the revised version of the manuscript we added a paragraph to the conclusions emphasizing that we see our study as an intermediate step towards a spatially resolved assessment: "We document large systematic differences in $R_{HgC}$ driven by (1) different contributions of geogenic Hg in mineral soils, e.g. resulting in higher $R_{HgC}$ in Alaska than in other areas of the northern circumpolar permafrost region and (2) the stability of Hg with respect to reemission from organic soils, e.g. resulting in a gradient with increasing $R_{HgC}$ towards the north of the WSL. These systematic differences illustrate the limitations of the poolsize estimation approach where C inventories are multiplied with average $R_{HgC}$ values and emphasize the need for spatially resolved sampling and pool size estimates, similar to the northern circumpolar permafrost C pool estimates (Hugelius et al. 2014). In particular, to estimate the release of Hg to aquatic ecosystems, e.g. coastal erosion and transfer to rivers, and Hg evasion to the atmosphere, spatially resolved Hg soil pools will be valuable."

Concerning western Siberia, where our sampling was performed, we are rather confident that the peat cores are representative and cover full territory needed for assessment of regional Hg pool. To further address this issue, we estimated the lateral variability in trace (toxic) metal concentration in peat cores from various micro-landscapes (mound, depression) in the same permafrost zone (latitude) based on our former work of elementary composition of peat across the WSL (Stepanova et al., 2015, Appl. Geochemistry, 53, 53–70, doi:10.1016/j.apgeochem.2014.12.004, Fig. 5, and associated Supplementary Information). These variations for Cd and Pb concentrations range from 25 to 50%. Similar range is exhibited by Fe and P. Although these elements cannot serve as straightforward analogues to Hg, we believe that the lateral variations in Hg concentration should be within the IQ range of Hg:C ratio as depicted in Fig. 4 of our manuscript, and as such, these variations do not sizably affect the overall estimation of Hg pools in Eurasian peat soils.

Furthermore, we also compared our unpublished data on elementary composition of peat with those of Raudina et al. (2019) and Stepanova et al. (2015) for the same key areas. We used a non-parametric Mann-Whitney U test for paired data at a significance level of 0.05 to assess the difference between sites (micro-landscapes) for each key area. The overwhelming majority of elements do not exhibit statistically significant differences between different peat cores. Thus, in the middle taiga region (Mukhrino), only Gd and Tb were sizably different. In Khanymey, only Mg showed statistically significant difference between different peat cores. In the forest tundra of Pangody, only Na, Ti, As, Cd, Tl, Pb exhibited sizable differences. Finally, various peat cores from southern tundra (Tazovsky) differed only in the concentration of Ca, Ni, Cu, Mo and Hf.

According to our results on 6 peat cores in this study, in the peat active layer, Hg was positively correlated with K, Rb, Cs, P, As, V, Cr, Cu (Table S3 of the manuscript). None of these elements demonstrated significant (at $p < 0.05$) differences between different microlandscapes and peat cores in Mukhrino and Khanymey. In the frozen part of the peat core, Hg was positively correlated with Ca, N, Mn, Sr, Mg, P (Table S3 of the manuscript). Except Ca, none of these elements differed between frozen parts of peat cores in 4 study sites of Raudina et al. (2019).

Taken together, we believe that one single core is 95% representative for the inter micro-landscapes variations and can adequately serve the purpose of assessment of both elementary composition and overall stock of elements (including Hg). We changed our wording in the text to represent this discussion (line 156): "The physical, chemical and botanical properties of several peat cores collected in the homogeneous palsa region in the north and ridgre-ryam complex in the south are highly similar among different peat mounds, suggesting that the cores we obtained are representative for the WSL (Velichko et al., 2011; Stepanova et al., 2015)."

**Table R1.** Comparison of concentrations of major and trace elements in peat cores from different microlandscapes (mound, depression) of WSL peatbogs studied in this work (and our unpublished data) with results of Raudina et al. (2019); Stepanova et al. (2015). Only the elements exhibiting statistically significant differences are presented.

| Elements | U | Z | p-value |
|---|---|---|---|
| Stepanova et al., 2015 Mukhrino, Middle taiga | | | |
| Gd | 6 | -3.3 | 0.001 |
| Tb | 0 | 3.5 | 0.000 |
| Our unpublished data Khanymey, Northern taiga | | | |
| Mg | 19 | 2.6 | 0.009 |
| Stepanova et al., 2015 Pangody, Forest tundra | | | |
| Na | 0 | 2.4 | 0.016 |
| Ti | 2 | -2.0 | 0.042 |
| As | 1 | -2.2 | 0.027 |
| Cd | 0 | -2.4 | 0.016 |
| Tl | 1 | -2.2 | 0.027 |
| Pb | 2 | -2.0 | 0.042 |
| Tazovsky, Pangody, Khanymey, Kogalym, Raudina et al., 2019 | | | |
| Ca | 20 | 2.4 | 0.017 |
| Ni | 2 | 3.7 | 0.000 |
| Cu | 22 | 2.2 | 0.025 |
| Mo | 17 | 2.6 | 0.009 |
| Hf | 16 | 2.7 | 0.008 |

References:

Raudina, T. V. and Loiko, S. V.: Properties and major element concentrations in peat profiles of the polygonal frozen bog in Western Siberia, in: IOP Conference Series: Earth and Environmental Science (Vol. 400, No. 1, p. 012009). IOP Publishing. (2019, November).

Stepanova, V. A., Pokrovsky, O. S., Viers, J., Mironycheva-Tokareva, N. P., Kosykh, N. P., and Vishnyakova, E. K.: Elemental composition of peat profiles in western Siberia: Effect of the micro-landscape, latitude position and permafrost coverage, Appl. Geochemistry, 53, 53–70, doi:10.1016/j.apgeochem.2014.12.004, 2015.

**Comment 4.**
**According to section 2.2, C pools were multiplied with the respective Hg:C ratios for organic and mineral soils from north America (excluding Alaska) and Eurasia to estimate the northern soil Hg pool. I am not entirely sure what the authors mean by "excluding Alaska". Did they estimate the northern soil Hg pool by applying different Hg:C ratios for Alaska, or by simply assuming Alaska does not exist? Please clarify.**

We acknowledge that Alaska still exists! However, we did not include Alaskan mineral soil Hg:C ratios in our estimate of the mineral soil Hg:C ratio representative for the entire northern circumpolar permafrost region. The reason is that the elevated Alaskan mineral soil Hg:C ratio is biased high and not representative of the large Siberian mineral soils. On L 360-362 of the discussion paper we quantify the systematic error made by "excluding Alaska": "The error made by neglecting high $R_{HgC}$ in Alaskan mineral soils is small, on the order of 2.5 Gg Hg, as estimated from the relatively small Alaskan C pool of 2.6 Pg C (Tarnocai et al., 2009)." The 0-3 m Hg pool in the northern circumpolar permafrost region is 557 Gg with an interquartile range between 371 and 699 Gg. We therefore argue that a systematic underestimation in the order of 2.5 Gg (approx. 0.5% of the total pool) is negligible given the large uncertainties associated with the estimate. The carbon pool estimates for different soil types from Hugelius et al (2014) are for the entire northern circumpolar permafrost region and no such data are available for Alaska on a soil type level. Therefore we are not able to provide a more accurate estimate at this stage.

**Comment 5.**
**Page 14 and Figure 8, the authors suggest that "North American and Eurasian mineral soils Hg:C ratio was lower than Hg:C ratio reported for Alaska". Additionally, "the Hg:C ratio in organic soils was approximately 4 times lower than that in mineral soils of North America and Eurasia". I do not understand which dataset was used here. I would appreciate a table with the list of studies the authors are referring to. In lines 345-346 the authors mention "the literature data compilations of Olson et al. (2018) and Schuster et al. (2018)" but this is to my point of view not enough.**

For our analysis we combine four datasets: the original permafrost Hg data from Schuster et al. 2018 (ca. 590 datapoints), a global compilation of Hg soil data by Schuster et al (ca. 11000 datapoints) a dataset of permafrost-affected Arctic and Boreal Hg soil data used by Olson et al. 2018 (958 datapoints) and the original soil data from the western Siberian lowlands (223 datapoints). We refer to the data availability statement, where we provide a link to the data sources used in this analysis. The dataset original in this study is provided in the supporting information of this study. The dataset used in Olson et al 2018 is currently not publically available and has to be acquired by contacting the corresponding author. We also added a new table (Table S4) in the Supporting Information providing an overview of the data used for the pan-Arctic assessment of $R_{HgC}$.

**Comment 6.**
**Same comment for the Hg:C ratios in various climate zones: which data were used? Again, I would really appreciate a table summarizing the literature used here. This entire section is too confusing as is.**

We refer to the data availability statement, where we provide a link to the data sources used in this analysis.

Data availability. Hg and C concentration data of the WSL soil samples are available in the supplement. The permafrost data from Schuster et al. 2018 and a global compilation of RHgC data is available as supplementary information (https://agupubs.onlinelibrary.wiley.com/doi/full/10.1002/2017GL075571, last access: 6 December 2019). The Arctic and boreal soil data from the Olson et al. 2018 study is available from the corresponding author upon request. Note that Olson et al. Table 1 has an incorrect organic soil $R_{HgC}$ of 0.274 Gg Pg$^{-1}$, which should be 0.119 Gg Pg$^{-1}$. The data from the tropical climate zone was compiled from original publications of Almeida (2005); Almeida et al. (2005); Campbell et al. (2003); Melendez-Perez et al. (2014).

We now added these reference to the main text as well.

**Comment 7.**
**The authors compare their 1084 Gg estimate of global Hg soil pool (0-30 cm) to the available literature. However, as mentioned by Outridge et al. (2018) (that should be cited here), most of these studies refer the amount of Hg in the actively recycling soil pool. For instance, the 950 Mg estimate by Outridge et al. (2018) refers to the top 10 cm. Similarly, Selin et al. (2008) referred to a layer _ 15 cm deep.**

We agree that the depth intervals to estimate the Hg pools in soils varies between different studies and thereby contributes to the large range in pool estimates. In the revised MS (Tables1, 2 and 3) we report the soil depth intervals of individual studies. The depth of 0 to 30 cm has been used in our study because this interval is established in the carbon community and soil carbon inventories exist for this depth range. We do not interpret this depth interval as the soil Hg pool that is actively recycled. Such a simplification would not take into account the heterogeneity between different soil types and the complexity of Hg cycling in soils. We added the Outridge et al. 2018 reference, but we could not find where the 950 Gg soil pool estimate for the 0-10cm is referred to. In Table 1 of the Outridge et al. (2018) paper, the soil Hg pool (described as organic layers) is estimated to be 150 Gg, but no depth interval is given in their work.

**Line-by-line comments:**

**Lines 38-39: "Hg concentrations increase from south to north in all soil horizons, reflecting enhanced net accumulation of atmospheric gaseous Hg by the vegetation Hg pump". As is, this sentence seems to suggest increasing vegetation uptake from south to north. However, as discussed in the manuscript, the Hg concentration increase is actually due to decreasing reemissions from south to north. Please edit this sentence accordingly (misleading as is).**

Thanks for pointing out this inconsistency. We changed the phrase as follows: "Hg concentrations increase from south to north in all soil horizons, reflecting a higher stability of sequestered Hg with respect to re-emission."

**Lines 70-71: see major comment #1.** We agree and changed the nomenclature accordingly.

**Line 82: "strong year round net Hg(0) emission". Please clarify what you mean by "strong".**
We deleted the word 'strong'

**Line 91: "GIS" please define acronym.**
Deleted the term 'GIS'; it is not critical; expanding would lengthen the phrase unnecessarily.

**Line 95: see major comment #2.**
We note that Olson et al. Table 1 has an incorrect organic soil $R_{HgC}$ of 0.274 Gg $Pg^{-1}$, which should be 0.119 Gg $Pg^{-1}$; the typo did not affect their soil Hg budgets."

**Line 126: please replace "atmospheric" by "ambient" and "increases" by "decreases".**
Changed as suggested

**Line 131: referring to the active layer as "unfrozen" soils is somewhat misleading since the active layer thaws during summer but freezes again in winter.**
We deleted the word "unfrozen"

**Lines 152-155: see major comment #3.**
We explained that field logistics and financial support did not make it possible to study multiple cores from each climate zone. Note that in terms of Hg concentration data points, the new WSL soil sample dataset represents only for 2% of the entire database used to assess the Hg soil pool and $R_{HgC}$ (Table 4, Fig. 9).

**Lines 166-177: please define acronyms (BCR, MESS, NIST, SRM, ICP-MS).**
It is fairly uncommon to fully write out acronyms of reference materials BCR, MESS, NIST SRM, i.e. doing a google on the acronyms (with reference number) will lead to the right information; doing a google on full terms will not. We expanded the term ICP-MS. SRM was deleted.

**Lines 187-189: unclear, see major comment #4.**
-The carbon pool estimates for different soil types from Hugelius et al (2014) are for the entire northern circumpolar permafrost region and no such data are available for Alaska on a soil type level.

**Line 190: typo, "singe" should be "single".**
Typo corrected

**Line 211 and throughout the manuscript: please use "PI" instead of the full name to make it easier to find the associated figure (same comment applies to all the sites).**
We would like to point out that this would add another 6 acronyms to the MS; we prefer to discuss sites by naming them fully; Note that the Figure captions include both full names and abbreviation, such as Plotnikovo (Pl).

**Lines 218-224: how does this compare to other studies? Please strengthen the discussion.**

Please note that we separated the results and discussion part in this manuscript. In section 3.1. we present the results from the peat cores sampled along the western Siberian lowlands. A comparison with other studies is given in the discussion section 4.2.

**Lines 225-229: how does this compare to other studies? Please strengthen the discussion.**
In section 3.1, we present the results from the peat cores sampled along the western Siberian lowlands. A comparison with other studies is provided in section 4.2.

**Line 301: for consistency please use the same units throughout the manuscript (Gg/Pg).**
We agree and corrected throughout the MS

**Line 320: see major comment #2.**
Necessary edits and corrections were applied

**Lines 322-323: see major comment #1.**
We agree and changed the nomenclature accordingly.

**Lines 328-329: please add units for the medians.**
The units were added

**Lines 318-352: I find this entire section confusing because I do not understand which data you are referring to. See major comment #5.**
We clarified as much as possible, adding "Dalton Highway, Noatak National Preserve, 8 Mile Lake Observatory" localities to the Olson et al., site description. We hope that after addition of new Table S4, the description is more clear.

**Lines 365-367: Please clarify which studies you are referring to. See major comment #6.**
We added references from the Data availability Statement to the main text:" (Campbell et al., 2003; Almeida, 2005; Almeida et al. 2005; Melendez-Perez et al., 2014; Olson et al., 2018)"

**Lines 369-373: See major comment #7.**
We explained specific soil depths with relevant references.

**Figure 3: the caption should be self-explanatory. What do ALT, PF1 and PF2 mean?**
The ALT stands for Active Layer Thickness and PF1 and PF2 designate surface and deep permafrost layers. See Table 1 for exact abbreviations of ALT, PF1 and PF2. Specifically, PF1 is frozen peat, (ALT-100 см); PF2 is frozen peat (ALT to mineral layer). We added a pertinent reference to Table 1, where these abbreviations are presented.

We thank both reviewers for very insightful and constructive comments. Care of these and other small self-motivated corrections, we believe the manuscript can meet high standards of the journal.

Looking forward to hearing from you

Yours sincerely
Oleg S. Pokrovsky, on behalf of co-authors